# ADAPTIVE TESTING FOR LLM EVALUATION: A PSYCHOMETRIC ALTERNATIVE TO STATIC BENCHMARKS

## ABSTRACT

Evaluating large language models (LLMs) typically requires thousands of benchmark items, making the process expensive, slow, and increasingly impractical at scale. Existing evaluation protocols rely on average accuracy over fixed item sets, treating all items as equally informative despite substantial variation in difficulty and discrimination. We introduce ATLAS, an adaptive testing framework based on Item Response Theory (IRT) that estimates model ability using Fisher information–guided item selection. ATLAS reduces the number of required items by up to 90% while maintaining measurement precision. For instance, it matches whole-bank ability estimates using only 41 items (0.157 MAE) on HellaSwag (5,600 items). We further reconstruct accuracy from ATLAS's ability estimates and find that reconstructed accuracies closely match raw accuracies across all five benchmarks, indicating that ability $\theta$ preserves the global performance structure. At the same time, $\theta$ provides finer discrimination within accuracy-equivalent models: among more than 3,000 evaluated models, 23–31% shift by more than 10 rank positions, and models with identical accuracies receive meaningfully different ability estimates. Code and calibrated item banks available at https://anonymous.4open.science/r/ATLAS-3210/README.md.

## 1 INTRODUCTION

Large language model evaluation relies on benchmarks with tens of thousands of items, which are costly to run and often take days or weeks to complete. Even with benchmarks exceeding 100,000 items, evaluation still depends on average accuracy over fixed item sets. This practice overlooks valuable statistical information and raises concerns about efficiency and validity.

Current evaluation practices face three fundamental limitations. First, average benchmark scores obscure meaningful differences between models with distinct error patterns, especially among lower-performing models where small ability differences are dominated by measurement noise. Second, static evaluations treat poorly discriminative items as equally informative as high-quality questions, leading to unreliable and often misleading comparisons. Third, evaluating complete benchmarks is inefficient and time-consuming, requiring models to answer hundreds or thousands of items regardless of how much additional information those items provide.

To address these limitations, we propose ATLAS (Adaptive Testing for LLM Ability Scoring), an adaptive evaluation framework based on computerized adaptive testing (CAT) (Lord, 1980; Wainer et al., 2000; Weiss, 1982). ATLAS first calibrates benchmark items using three-parameter logistic (3PL) IRT models to estimate item difficulty, discrimination, and guessing parameters (Birnbaum, 1968; Hambleton et al., 1991). Then, rather than administering fixed item sets, ATLAS dynamically selects items with maximum Fisher information for each model's current estimated ability, terminating when precision thresholds are reached. This approach directly addresses all three limitations: Fisher information-guided selection provides precise ability estimates that distinguish models with identical accuracy, dynamic item selection prioritizes highly discriminative items rather than treating all questions as equally informative, and adaptive termination enables reliable evaluation with far fewer items and substantially less time than full-benchmark scoring.

We evaluate ATLAS across five major benchmarks, including WinoGrande (Sakaguchi et al., 2021), TruthfulQA (Lin et al., 2021), HellaSwag (Zellers et al., 2019), GSM8K (Cobbe et al., 2021), ARC (Clark et al., 2018) and find that it matches or exceeds the accuracy of strong static baselines while

using far fewer items. For example, ATLAS achieves the lowest MAE on TruthfulQA (0.064 with 48 items) and HellaSwag (0.157 with 41 items), and matches MetaBench (Kipnis et al., 2025) on WinoGrande while using 2× fewer items (70 vs. 133). It also outperforms TinyBenchmarks (Polo et al., 2024), which uses 97–100 items but yields higher error across all benchmarks. Overall, ATLAS requires only 30–89 items per benchmark compared to hundreds in static subsets, and attains the lowest Information Efficiency Score (IES) across all benchmarks, demonstrating the strongest accuracy–efficiency tradeoff.

Our contributions are: (1) We identify fundamental limitations of average-score evaluation and show that psychometric ability estimates provide more robust and informative comparisons of LLM performance. (2) We introduce ATLAS, a large-scale adaptive testing framework for LLMs that achieves up to 90% item reduction while maintaining measurement precision through SE-controlled stopping, enabling flexible and precision-targeted evaluation beyond fixed-length designs. (3) We conduct a comprehensive psychometric analysis of five major benchmarks, revealing that IRT-based ability estimation induces substantial rank reordering (23–31% of models shift by more than 10 positions). (4) We highlight the importance of rigorous psychometric validation by reporting model-fit statistics (e.g., RMSEA via the M2 statistic) and demonstrating the use of common-person linking to align item parameters efficiently and ensure cross-model comparability.

## 2 RELATED WORK

### 2.1 IRT-BASED APROACHES

Item Response Theory (IRT) has recently been applied to LLM evaluation (Lalor et al., 2024; Guinet et al., 2025). It provides item parameters such as difficulty, discrimination, and guessing, as well as latent ability estimates $\theta$ for models. However, existing IRT applications remain largely **static** in nature. For instance, TinyBenchmarks (Polo et al., 2024) uses clustering for item selection but doesn't guarantee informativeness for $\theta$ estimation, while MetaBench (Kipnis et al., 2025) requires computationally expensive iterations to identify stable subsets. Moreover, these approaches often lack proper psychometric validation and emphasize predictive accuracy over **model fit**. TinyBenchmarks and MetaBench do not report fit statistics. Instead, we ccompute these metrics using their released IRT code (as shown in Table 1). This limitation makes it difficult to ensure that the resulting ability estimates are valid, interpretable, and comparable across models. A detailed comparison of IRT-based approaches is provided in Appendix B.

Beyond these limitations of existing IRT applications, many evaluations continue to rely on average scores. Average scores tend to mask meaningful model differences and are often affected by form-dependence, nonlinear scaling, equal weighting of uninformative items, and contamination sensitivity (see Appendix A for detailed analysis). In contrast, IRT-based ability estimates ($\theta$) provide form-invariant, uncertainty-aware alternatives that adjust for item difficulty and discrimination.

### 2.2 ADAPTIVE TESTING

Computerized adaptive testing (CAT) adjusts item administration based on an examinee's evolving ability estimate (Meijer & Nering, 1999; Van der Linden & Glas, 2010). After each response, the test updates the ability estimate and selects the next item using an algorithm that aims to provide the most informative measurement while satisfying test constraints (Weiss, 1982; Chang, 2015; Cheng & Chang, 2009). Related adaptive frameworks such as multistage testing and process-data–based approaches apply similar principles and offer additional flexibility and diagnostic information (Zenisky et al., 2009; Zheng & Chang, 2015; Tang et al., 2024). These features allow CAT to evaluate examinees efficiently while maintaining rigorous measurement precision. This adaptive structure aligns well with challenges in evaluating LLMs, which vary widely in their performance levels. Current evaluations often use large static benchmarks in which every model must answer all items, even when many items provide little information about its ability. These benchmarks also rarely report empirical item characteristics, so their difficulty range and informativeness across models remain unclear. CAT addresses these limitations by selecting items targeted to each model's estimated ability, which yields more precise and efficient evaluation with far fewer items.

However, only a few studies have explored adaptive evaluation for LLMs. Early efforts were either limited in scope (Zhuang et al., 2023) or primarily conceptual (Zhuang et al., 2025). A recent study

that is closely related to our work is Fluid Benchmarking (Hofmann et al., 2025), which appeared around the same time as this research. Fluid also applies CAT principles to increase evaluation efficiency and models LLM performance on a latent ability scale. The two approaches are complementary and differ in several ways. Fluid focuses on adaptive evaluation during LLM pretraining, whereas our work examines post-training evaluation. Fluid calibrates its IRT model on 102 LMs, while ATLAS uses a substantially larger and more diverse pool of 3,000+ LMs. Fluid adopts a fixed-length adaptive design, while ATLAS uses a precision-based stopping rule that terminates the test once the uncertainty of the ability estimate falls below a predefined threshold. Precision-based stopping ensures consistent measurement precision across models and avoids administering unnecessary items. In addition, we report model-fit statistics to ensure the adequacy of the IRT model before running CAT and provide a transparent description of calibration and linking procedures used to estimate item parameters. Both studies demonstrate that adaptive testing methods can be used at different stages of LLM development and under different design choices, which illustrates the broader potential of CAT-based approaches for scalable and precise LLM assessment.

## 3 METHODOLOGY

We introduce a novel adaptive testing framework that transforms LLM evaluation from static benchmarking to dynamic ability estimation. Our approach addresses three critical limitations of current evaluation practice: (1) it reduces computational cost by requiring 90% fewer items while maintaining accuracy, (2) it overcomes the ceiling effects of accuracy-based metrics and preserves discrimination across the ability spectrum, and (3) it distinguishes models with identical average scores but different underlying capability patterns.

This section presents our framework in four stages: problem formulation (Section 3.1), data construction with psychometric filtering (Section 3.2), item bank calibration using IRT models (Section 3.3), and adaptive testing with randomesque selection (Section 3.4).

### 3.1 PROBLEM FORMULATION AND SETUP

We formulate LLM evaluation as a psychometric measurement problem. Let $\mathcal{I}$ denote the set of benchmark items and $\mathcal{L}$ the set of language models. For each model $\ell \in \mathcal{L}$ and item $i \in \mathcal{I}$, we observe a binary response $Y_{i,\ell} \in \{0, 1\}$, where 1 indicates correct and 0 incorrect. These responses form the item-response matrix $\{Y_{i,\ell}\}_{i \in \mathcal{I}, \ell \in \mathcal{L}}$.

Unlike traditional approaches that rely solely on accuracy scores, our objective is to estimate the latent ability $\theta_\ell$ of each model based on its response pattern $\{Y_{i,\ell}\}_{i \in \mathcal{I}}$, while simultaneously calibrating item-level parameters: discrimination $a_i$, difficulty $b_i$, and guessing $c_i$. This approach enables fine-grained model comparison even when models achieve identical accuracy, as $\theta_\ell$ accounts for the varying informativeness of different items.

### 3.2 DATA CONSTRUCTION WITH PSYCHOMETRIC FILTERING

We construct the item-response matrix using data from the HuggingFace Open LLM Leaderboard. The item pool $\mathcal{I}$ spans five benchmarks: ARC, GSM8K, HellaSwag, TruthfulQA, and WinoGrande. To ensure data quality for IRT calibration, we apply two levels of filtering: removing unsuitable models and eliminating non-informative items.

**Model Selection and Splitting.** We retain only models $\mathcal{L}$ with complete responses across all items. To obtain a calibration sample whose ability distribution approximates a Gaussian, a standard assumption for stable IRT estimation, we exclude models in the extreme low-ability tail (below the 0.1st percentile), whose near-zero response patterns destabilize 3PL parameter estimation. The high-ability tail is small and non-degenerate, so these models are retained. The selected models are then split into training and testing sets using stratified random sampling (10 bins) to ensure that both splits share a similar ability distribution. We allocate 90% of the models to the training set for item calibration, and use the remaining 10% as the testing set for evaluating performance in our experiments (see Table 5).

**Item Filtering.** We apply two complementary filters to retain only discriminative items:

- **Low-variance removal:** Items with response standard deviation $< 1\%$ or mean accuracy $> 95\%$ are discarded, as they provide litter information for differentiating between models.

- **Discrimination filtering:** We compute the point-biserial correlation $r_{pb}(i)$ between each item's response vector $\{Y_{i,\ell}\}_{\ell \in \mathcal{L}}$ and the models' total scores $T_\ell = \sum_{j \in \mathcal{I}} Y_{j,\ell}$ (see Appendix C for details). Items with $r_{pb}(i) < 0.1$ are removed as non-diagnostic.

This filtering process yields a refined response matrix that supports stable and reliable IRT calibration (see Table 5 in Appendix C for detailed results).

## 3.3 SCALABLE IRT CALIBRATION

The calibration stage estimates item parameters $(a_i, b_i, c_i)$ and computes reference ability estimates $\hat{\theta}_\ell^{\text{whole}}$ for each LLM $\ell$ for validation. To model the probability of a correct response, we adopt the three-parameter logistic (3PL) IRT model (Birnbaum, 1968; Lord, 1980):

$$p_i(\theta_\ell) = c_i + \frac{1 - c_i}{1 + \exp(-a_i(\theta_\ell - b_i))}. \tag{1}$$

Here, $a_i$ is the discrimination parameter, which determines how sharply item $i$ differentiates between stronger and weaker models. $b_i$ is the difficulty parameter, specifying the ability level at which a model has a 50% chance (beyond guessing) of answering item $i$ correctly. $c_i$ is the guessing parameter, setting the lower bound on the probability of success due to random guessing. These parameters enable Fisher information-based prioritization of items in our adaptive framework, distinguishing high-quality items from those with low discriminative power.

**Common-Person Calibration at Scale.** To estimate item characteristics efficiently using the 3PL model, we adopted a partition-based calibration procedure that leverages the unique structure of LLM benchmarking. Instead of fitting the full 3PL model to the entire item pool at once, which would be computationally prohibitive, we divided the items into $K$ non-overlapping subsets $\mathcal{I}_k$ (each with $|\mathcal{I}_k| \geq 100$ items), and calibrated each subset independently. This yields multiple provisional difficulty scales that must be aligned. Because all models answer all items, the model population serves as a natural set of common persons, allowing us to link the independently calibrated subsets onto a unified scale using mean–sigma transformations (Kolen & Brennan, 2014). This approach reduces computational complexity from $O(|\mathcal{I}|^3)$ to $O(K \cdot \max_k |\mathcal{I}_k|^3)$ while maintaining calibration accuracy due to the stability provided by having all models serve as linking anchors (Chalmers, 2012). A detailed description of this common-person calibration and linking procedure is provided in Appendix C.2.

**Heterogeneity-Aware Ability Estimation.** LLM populations exhibit extreme heterogeneity, ranging from near-random models ($\theta \approx -3$) to highly capable systems ($\theta \approx 3$). To obtain stable and unbiased estimates across this wide ability spectrum, we adopt the Weighted Likelihood Estimator (WLE) (Warm, 1989), which incorporates a bias-correction term $\frac{J(\theta)}{2I(\theta)}$, where $J(\theta) = \sum_i \frac{\partial I_i(\theta)}{\partial \theta}$. WLE provides finite, well-behaved estimates even at ability extremes and maintains desirable consistency properties (Baker & Kim, 2004). These characteristics are essential for establishing reliable evaluation baselines under the substantial heterogeneity present in modern LLM benchmarks.

**Multi-Subset Model Fit Validation.** Unlike prior IRT applications to LLM evaluation (Polo et al., 2024; Kipnis et al., 2025), which do not report any model-fit diagnostics, we conduct rigorous psychometric validation to ensure calibration quality. We compute the limited-information $M_2$ statistic (Maydeu-Olivares, 2015) with RMSEA indices for TinyBenchmarks, MetaBench, and ATLAS (see Table 1). TinyBenchmarks exhibits extremely poor fit across all benchmarks. This is expected: TinyBenchmarks is calibrated on a relatively small set of 395 LLMs while relying on IRT models with up to 15 latent traits, creating a parameter space that far exceeds the available data. Such an underidentified setting makes good model fit difficult, and our computed RMSEA values confirm severe misfit. MetaBench performs better, with RMSEA values between 0.04 and 0.14, yet still shows poor fit on TruthfulQA (0.1389) and marginal fit on ARC (0.0811), indicating that its fixed subsets do not generalize evenly across datasets. In contrast, ATLAS consistently achieves acceptable or good fit across all benchmarks, demonstrating the stability and robustness of our calibration procedure.

Besides, our partition-based calibration strategy, which divides the full item bank into non-overlapping subsets (each containing $\geq 100$ items for statistical stability), enables both computational feasibility and robust validation. Since the same set of models $\mathcal{L}$ acts as common persons across all partitions, diagnostic statistics reflect global calibration quality rather than partition-specific artifacts. This multi-subset linking design ensures that model fit metrics capture systematic patterns across the entire item bank, not just localized subsets. This validation is crucial for reliable adaptive testing, as misfitting items would compromise Fisher information calculations and degrade selection accuracy.

Table 1: Model fit comparison across benchmarks using the limited-information statistic $M_2$ and its derived Avg. RMSEA values. Lower Avg. RMSEA indicates better model fit. Model fit is interpreted according to standard psychometric thresholds: *RMSEA < 0.05 = Good fit; 0.05–0.08 = Acceptable fit; 0.08–0.10 = Marginal fit; > 0.10 = Poor fit.*

| Method | Winogrande | | TruthfulQA | | HellaSwag | | GSM8K | | ARC | |
|---|---|---|---|---|---|---|---|---|---|---|
| | RMSEA | Fit | RMSEA | Fit | RMSEA | Fit | RMSEA | Fit | RMSEA | Fit |
| TinyBenchmarks | 364.24 | Poor | 371.49 | Poor | 646.82 | Poor | 506.60 | Poor | 369.89 | Poor |
| MetaBench | 0.0524 | Acceptable | 0.1389 | Poor | 0.0498 | Good | 0.0423 | Good | 0.0811 | Marginal |
| ATLAS | 0.0565 | Acceptable | 0.0690 | Acceptable | 0.0482 | Good | 0.0438 | Good | 0.0595 | Acceptable |

## 3.4 ADAPTIVE TESTING WITH INFORMATION SELECTION

Our proposed ATLAS dynamically selects the most informative items for each model, dramatically reducing the number of items needed while maintaining accuracy. Algorithm 1 presents the complete procedure. The algorithm includes several key design choices tailored to LLM evaluation:

---

**Algorithm 1** Adaptive Testing for Model $\ell$

---

1: **Initialize:** $\hat{\theta}_0 \leftarrow 0$, test record $R_\ell \leftarrow \emptyset$, $t \leftarrow 0$
2: **while** $t < \text{max\_items}$ and not converged **do**
3:      $t \leftarrow t + 1$
4:      **if** $t = 1$ **then**
5:          Select item $i_t$ with $|b_{i_t} - \hat{\theta}_0|$ minimized
6:      **else**
7:          Compute Fisher information $I_i(\hat{\theta}_{t-1})$ for all unadministered items
8:          Select $i_t$ randomly from top-5 most informative items
9:      **end if**
10:      Administer item $i_t$ to model $\ell$, observe response $Y_{i_t,\ell}$
11:      Update record: $R_\ell \leftarrow R_\ell \cup \{(i_t, Y_{i_t,\ell})\}$
12:      Update ability: $\hat{\theta}_t \leftarrow \text{EAP}(R_\ell)$
13:      Compute standard error: $\text{SE}(\hat{\theta}_t) \leftarrow 1/\sqrt{\sum_{j \in R_\ell} I_j(\hat{\theta}_t)}$
14:      **if** $t \geq \text{min\_items}$ and $\text{SE}(\hat{\theta}_t) \leq \tau$ **then**
15:          **break**             ▷ Convergence achieved
16:      **end if**
17: **end while**
18: **return** $\hat{\theta}_\ell \leftarrow \hat{\theta}_t$, $\text{SE}(\hat{\theta}_\ell)$, $R_\ell$

---

**Initialization and Bounds.** We initialize the ability estimate at $\hat{\theta}_0 = 0$. This value is a conventional neutral starting point in CAT because the latent trait scale is typically assumed to be centered at zero. Beginning at the scale midpoint helps stabilize early item selection by preventing the algorithm from drifting toward artificially high or low values before any response information is available. We enforce minimum (30) and maximum (500) item limits. The minimum ensures stable estimation for models at performance extremes, while the maximum constrains computational cost and yields approximately 90% reduction in test length relative to full benchmarks.

**Randomesque Item Selection.** Rather than deterministically selecting the single most informative item, we randomly sample from the top-5 candidates ranked by Fisher information:

$$I_i(\theta) = a_i^2 \cdot p_i(\theta) \cdot [1 - p_i(\theta)]. \tag{2}$$

This randomesque strategy (Kingsbury & Zara, 1989) prevents over-reliance on specific item types while still keeping high information, which is important for models with specialized capabilities.

**Sequential Ability Updates.** After each item administration, we update the ability estimate using Expected A Posteriori (EAP) estimation (Bock & Mislevy, 1982):

$$\hat{\theta}_t = \mathbb{E}[\theta|R_\ell] = \int \theta \cdot p(\theta|R_\ell) \, d\theta. \tag{3}$$

EAP provides numerically stable updates with sparse early responses and incorporates prior knowledge about ability distributions. In contrast, WLE tends to become unstable when response patterns are extreme, a situation common in the early stages of adaptive testing.

**Precision-Based Stopping.** In our implementation, testing stops once either the maximum item limit is reached or $\mathrm{SE}(\hat{\theta}_\ell)$ falls below a threshold $\tau$, after a minimum number of items has been administered. This precision-based rule ensures consistent measurement accuracy while minimizing test length. Although our experiments adopt this precision-based design, ATLAS can also operate under a fixed-length stopping rule by specifying a predetermined test length.

**Output and Validation.** For each model $\ell$, the algorithm produces: (1) the administered item sequence and responses $R_\ell$, (2) the ability estimate trajectory $\{\hat{\theta}_t\}$ with associated standard errors, and (3) the final estimate $\hat{\theta}_\ell$. We validate these adaptive estimates against whole-bank references $\hat{\theta}_\ell^{\mathrm{whole}}$ to confirm that our dramatic reduction in items does not compromise measurement accuracy.

## 4 EXPERIMENTS

We evaluate the proposed ATLAS framework across five benchmarks, comparing its efficiency and accuracy against static baselines such as random sampling, TinyBenchmarks, and MetaBench. We report accuracy- and efficiency-based metrics, with full metric definitions provided in Appendix E.

### 4.1 EXPERIMENTAL SETUP AND METRICS

We evaluate ATLAS across five diverse benchmarks covering different cognitive domains: Wino-Grande (commonsense reasoning), TruthfulQA (factual consistency), HellaSwag (procedural inference), GSM8K (mathematical reasoning), and ARC (scientific question answering). All experiments use calibrated item banks from Section 3.3.

We compare against four static baselines that do not adapt to individual models: (1) **Random sampling** of 100 items uniformly from the full bank, (2) **TinyBenchmarks** (Polo et al., 2024) using predetermined subsets selected via clustering without Fisher information optimization, (3) **MetaBench-Primary** and (4) **MetaBench-Secondary** (Kipnis et al., 2025) using curated splits that require computationally expensive iterations to identify stable subsets. Unlike these static approaches, ATLAS uses three precision thresholds ($\mathrm{SE}(\hat{\theta}) \leq 0.1, 0.2, 0.3$) and an item bound of 30–500, terminating adaptively when the required precision is achieved or the maximum test length is reached. Additional experimental configurations are provided in Appendix D.

We evaluate each method primarily in **ability space**, comparing ATLAS-estimated abilities $\hat{\theta}_\ell$ with full-bank abilities $\hat{\theta}_\ell^{\mathrm{whole}}$ (See Table 2). We report four metrics: (1) **Mean Absolute Error (MAE)** to measure estimation accuracy; (2) **Standard Error (SE)** of the absolute errors across models to quantify stability; (3) **Average Test Length**, the number of items administered per model; and (4) **Information Efficiency Score (IES)**, which jointly reflects accuracy and item usage relative to a 100-item random baseline (values $< 1$ indicate higher efficiency). Detailed formulations of these metrics are provided in Appendix E.

For completeness, we also provide **accuracy-space evaluations**, comparing reconstructed accuracies to full-bank raw accuracies (See Table 3). Additional evaluation metrics and their results, including **Item Exposure Rate**, **Test Overlap Rate**, and **Selection Time** are provided in Appendix G.1.

Table 2: Comparison of whole-bank ability $\hat{\theta}_\ell^{\text{whole}}$ and subset-based ability $\hat{\theta}_\ell$ across benchmarks. For each method, we report MAE±SE, item count, and Information Efficiency Score (IES), where lower values are better for all metrics. Bold indicates the best result, underlining the second-best, and dashed underlining the third-best.

| Method | WinoGrande | | | TruthfulQA | | | HellaSwag | | | GSM8K | | | ARC | | |
|---|---|---|---|---|---|---|---|---|---|---|---|---|---|---|---|
| | MAE±SE ↓ | Items ↓ | IES ↓ | MAE±SE ↓ | Items ↓ | IES ↓ | MAE±SE ↓ | Items ↓ | IES ↓ | MAE±SE ↓ | Items ↓ | IES ↓ | MAE±SE ↓ | Items ↓ | IES ↓ |
| Random$_{100}$ | 0.167±0.007 | 100 | 1.000 | 0.103±0.004 | 100 | 1.000 | 0.240±0.010 | 100 | 1.000 | 0.150±0.014 | 100 | 1.000 | 0.183±0.007 | 100 | 1.000 |
| TinyBenchmarks | 0.204±0.008 | 100 | 1.221 | 0.145±0.007 | 97 | 1.370 | 0.198±0.009 | 97 | 0.797 | 0.164±0.014 | 100 | 1.089 | 0.172±0.007 | 99 | 0.932 |
| MetaBench-P | **0.152±0.007** | 133 | 1.216 | 0.084±0.004 | 154 | 1.262 | 0.514±0.016 | 93 | 1.990 | 0.103±0.013 | 237 | 1.628 | 0.134±0.005 | 145 | 1.062 |
| MetaBench-S | 0.195±0.009 | 106 | 1.239 | 0.072±0.003 | 136 | 0.945 | 1.570±0.055 | 58 | 3.788 | **0.096±0.012** | 249 | 1.595 | 0.134±0.006 | 100 | 0.735 |
| ATLAS$_{0.1}$ | 0.155±0.012 | 70 | 0.655 | **0.064±0.002** | 48 | 0.300 | **0.157±0.010** | 41 | 0.266 | 0.150±0.011 | 70 | 0.701 | **0.084±0.006** | 89 | 0.407 |
| ATLAS$_{0.2}$ | 0.166±0.010 | 37 | 0.372 | 0.073±0.003 | 30 | 0.211 | 0.163±0.009 | 30 | 0.203 | 0.177±0.012 | 36 | 0.428 | 0.120±0.008 | 35 | 0.232 |
| ATLAS$_{0.3}$ | 0.179±0.011 | 32 | 0.342 | 0.071±0.003 | 30 | 0.206 | 0.165±0.010 | 30 | 0.205 | 0.173±0.012 | 31 | 0.363 | 0.117±0.007 | 30 | 0.193 |

Table 3: Comparison of raw whole-bank accuracy and p-IRT reconstructed accuracy across benchmarks. For each method, we report MAE±SE, number of administered items, and the Information Efficiency Score (IES), where lower values are better for all metrics. Bold denotes the best value, underlining the second-best, and dashed underlining the third-best.

| Method | WinoGrande | | | TruthfulQA | | | HellaSwag | | | GSM8K | | | ARC | | |
|---|---|---|---|---|---|---|---|---|---|---|---|---|---|---|---|
| | MAE±SE ↓ | Items ↓ | IES ↓ | MAE±SE ↓ | Items ↓ | IES ↓ | MAE±SE ↓ | Items ↓ | IES ↓ | MAE±SE ↓ | Items ↓ | IES ↓ | MAE±SE ↓ | Items ↓ | IES ↓ |
| Random$_{100}$ | 0.049±0.001 | 100 | 1.000 | 0.021±0.001 | 100 | 1.000 | 0.024±0.001 | 100 | 1.000 | 0.026±0.001 | 100 | 1.000 | 0.029±0.001 | 100 | 1.000 |
| TinyBenchmarks | 0.050±0.001 | 100 | 1.010 | 0.025±0.001 | 97 | 1.154 | **0.019±0.001** | 97 | 0.782 | 0.028±0.001 | 100 | 1.071 | 0.031±0.001 | 99 | 1.041 |
| MetaBench-P | 0.054±0.001 | 133 | 1.446 | 0.017±0.001 | 154 | 1.266 | 0.050±0.001 | 93 | 1.943 | 0.022±0.001 | 237 | 2.060 | **0.027±0.001** | 145 | 1.350 |
| MetaBench-S | 0.051±0.001 | 106 | 1.103 | 0.021±0.001 | 136 | 1.394 | 0.115±0.004 | 58 | 2.793 | **0.020±0.001** | 249 | 1.954 | 0.033±0.001 | 100 | 1.114 |
| ATLAS$_{0.1}$ | **0.048±0.001** | 70 | 0.678 | 0.023±0.001 | 48 | 0.532 | 0.020±0.001 | 41 | 0.348 | 0.039±0.001 | 70 | 1.055 | 0.032±0.002 | 89 | 0.974 |
| ATLAS$_{0.2}$ | 0.051±0.002 | 37 | 0.383 | 0.024±0.001 | 30 | 0.338 | 0.021±0.001 | 30 | 0.258 | 0.044±0.002 | 36 | 0.612 | 0.034±0.002 | 35 | 0.404 |
| ATLAS$_{0.3}$ | 0.050±0.001 | 32 | 0.324 | 0.023±0.001 | 30 | 0.331 | 0.021±0.001 | 30 | 0.261 | 0.042±0.002 | 31 | 0.516 | 0.034±0.002 | 30 | 0.350 |

## 4.2 PERFORMANCE AND RELIABILITY ANALYSIS

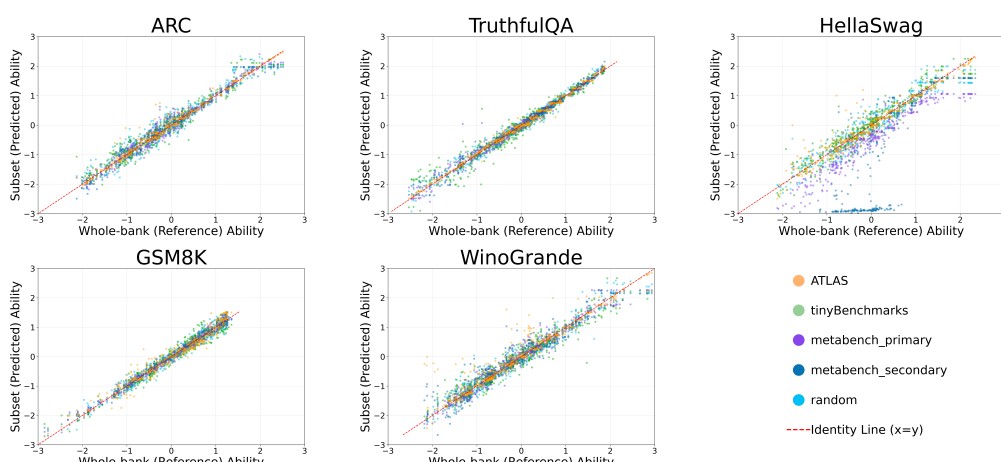

Figure 1: Comparison of subset (predicted) ability estimates against whole-bank (reference) abilities across five benchmarks (graphical illustration complementing Table 2). Points along the identity line indicate perfect agreement. ATLAS maintains the closest alignment overall, particularly on TruthfulQA, ARC and HellaSwag and in the high-ability regime of WinoGrande, while static baselines such as TinyBenchmarks and Metabench show greater variance and systematic deviation.

Table 2 presents a comparison of whole-bank ability estimates and subset-based estimates across five benchmarks. ATLAS consistently delivers the strongest accuracy–efficiency tradeoff among all methods. Across every benchmark, an ATLAS variant achieves the lowest Information Efficiency Score (IES), indicating that it provides the most accurate estimates using the fewest items. For example, ATLAS attains the best MAE on TruthfulQA (0.064 with 48 items) and HellaSwag (0.157 with 41 items), and matches the performance of MetaBench-Primary on WinoGrande while requiring nearly half as many items (70 vs. 133). Even in more challenging settings such as GSM8K and ARC, ATLAS maintains low MAE with item counts as small as 30–36, outperforming all static baselines in information efficiency. In contrast, static subsets such as TinyBenchmarks and MetaBench exhibit inconsistent performance. They are strong on some datasets but poor on others, with substantially higher IES values. Overall, these results demonstrate that adaptive item selection enables

ATLAS to deliver high-fidelity ability estimates while dramatically reducing evaluation cost, achieving reliable performance that fixed subsets fail to match.

Figure 1 plots subset-based ability estimates against whole-bank references across all five benchmarks. Across datasets, ATLAS shows the closest alignment to this identity line, with tightly clustered points and minimal systematic drift, reflecting stable and high-fidelity ability estimation. In contrast, static baselines exhibit benchmark-dependent failures. TinyBenchmarks consistently deviate from the identity line in the extreme high or low-ability regime, especially on ARC and TruthfulQA. MetaBench performs reasonably on GSM8K but breaks down substantially on HellaSwag, where both its primary and secondary subsets produce large downward deviations, indicating poor item coverage.

Table 6 shows that ATLAS produces diverse and efficient adaptive tests. Test overlap remains low (11–23%) and item exposure rates stay below 12% across all benchmarks, indicating broad item coverage rather than reliance on a small subset. Runtime is also practical, with end-to-end selection times ranging from 9.4 to 75.5 seconds per model, scaling predictably with bank size (fastest on TruthfulQA, longest on HellaSwag). Overall, ATLAS provides adaptive evaluations that are both statistically robust and computationally efficient.

**Accuracy Reconstruction.** To evaluate whether the ability estimates $\theta$ align with traditional accuracy-based evaluation, we reconstruct each model's expected accuracy from its estimated ability and its observed responses on the reduced subset using the *performance-IRT* (p-IRT) estimator (Polo et al., 2024) (detailed in Appendix F). Conceptually, p-IRT is grounded in the Test Characteristic Curve (TCC) (Lord & Novick, 2008; Hambleton et al., 1991), which maps a model's ability $\hat{\theta}$ to its expected probability of answering benchmark items correctly under the calibrated 3PL model. The p-IRT estimator refines this TCC-based mapping by combining the model's observed responses with IRT-predicted probabilities. Following this formulation, we convert each model's $\hat{\theta}$ into a reconstructed accuracy and compare it with the raw full-bank accuracy in Table 3. Across all five benchmarks, the reconstructed accuracies closely match the raw accuracies, indicating that $\theta$ preserves the global performance structure while providing finer discrimination than accuracy alone.

(a) GSM8K  (b) HellaSwag

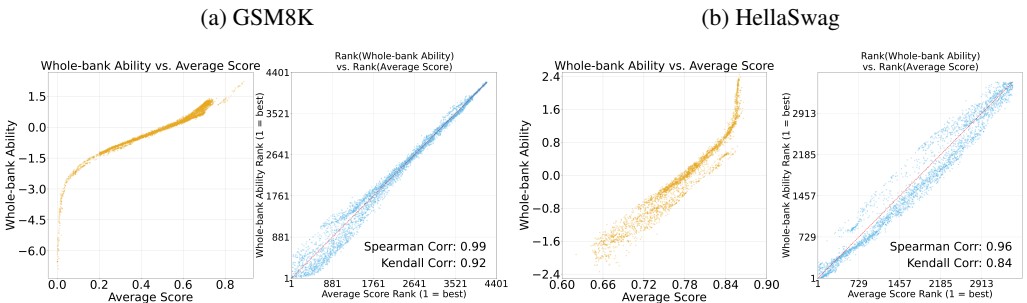

Figure 2: Comparison of IRT ability estimates $\hat{\theta}_\ell^{\text{whole}}$ with raw accuracy. Left: Ability vs accuracy reveals strong correlation but critical differences at performance extremes where accuracy collapses. Right: Rank comparison shows systematic reordering, with 23% (GSM8K) and 31% (HellaSwag) of models shifting $> 10$ positions. IRT separates models with identical accuracies by accounting for which items they solve correctly.

### 4.3 DISTINGUISHING LOW- AND HIGH-PERFORMING MODELS

Figure 2 demonstrates IRT's key advantage: separating models with similar accuracy scores through ability estimates. Despite strong correlations (0.99 for GSM8K, 0.96 for HellaSwag), systematic differences emerge at performance extremes. In low-performing regimes, accuracy collapses into narrow bands (0.10-0.15) while IRT spans $\theta \approx -3$ to $-1$, distinguishing models that solve easy versus challenging items. In high-performing regimes, ceiling effects compress accuracy differences, but IRT maintains discrimination across $\theta \approx 1.5$ to $2.5$. Similar patterns are observed across TruthfulQA, WinoGrande, and ARC benchmarks, with additional score-versus-theta comparisons provided in Appendix G.

The right panels show systematic rank reordering: 23-31% of models shift $> 10$ positions when ranked by IRT versus accuracy. Models performing well on hard items receive higher IRT ranks despite moderate accuracy, while those succeeding on easy items are appropriately downweighted. This enhanced discrimination provides more reliable model comparisons, especially critical in saturated performance regions where accuracy-based evaluation fails. Similar patterns of IRT superiority in distinguishing models are observed across TruthfulQA, WinoGrande, and ARC benchmarks, with additional analysis provided in Appendix G.

## 4.4 ITEMS ARE NOT EQUALLY INFORMATIVE

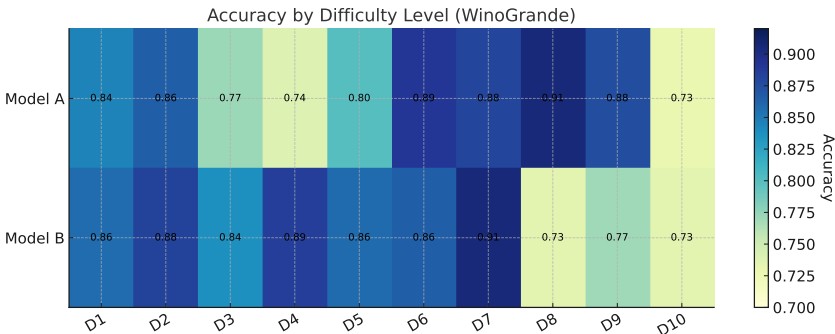

Figure 3: Two models with identical accuracy (0.833) on WinoGrande receive different ability estimates ($\hat{\theta}_A = 1.2$ vs $\hat{\theta}_B = 0.6$). Model A succeeds on harder items (darker cells on right), while Model B answers easier items (darker cells on left). IRT captures these item difficulty patterns that raw accuracy cannot.

Figure 3 demonstrates IRT's key advantage: models with identical accuracy (0.833) receive different ability estimates ($\hat{\theta}_A = 1.2$ vs $\hat{\theta}_B = 0.6$) based on which items they solve correctly. Under the 3PL IRT model, items differ in discrimination $a_i$, difficulty $b_i$, and guessing $c_i$ parameters. Model A succeeds on high-difficulty items ($b_i > 0.5$) with strong discrimination ($a_i > 1.5$), yielding 2.3× more Fisher information than Model B, which mainly answers easier, less discriminative items ($b_i < -0.5$, $a_i < 0.8$).

IRT provides automatic quality control by weighting items according to their empirical contribution to distinguishing model abilities. In our calibrated banks, 3.2-5.7% of items exhibit negative discrimination ($a_i < 0$), indicating systematic flaws where stronger models perform worse. For example, WinoGrande item #247 achieves 0.89 accuracy but $a_i = -0.43$ due to exploitable linguistic artifacts. Under raw accuracy, this flawed item contributes equally (weight = 1/N) to all scores, potentially inflating weak models. Under IRT, negative discrimination automatically down-weights its contribution by 82% (effective weight $\propto a_i^2 \approx 0.18$), reducing measurement contamination and providing more reliable ability estimates than accuracy alone. Similar patterns of identical accuracy leading to different ability estimates are observed across ARC, HellaSwag, and other benchmarks, with additional heatmap visualizations provided in Appendix G.

## 5 DISCUSSION

### 5.1 PSYCHOMETRIC CONSIDERATIONS FOR FUTURE LLM BENCHMARKS

Our findings highlight several important considerations for the construction of future LLM benchmarks. First, **item quality** directly affects evaluation reliability. Recent studies show that mislabeled or ambiguous items are common in existing LLM benchmarks (Vendrow et al., 2025; Gema et al., 2025). Although IRT naturally downweights such items through probabilistic modeling, static benchmarks lack mechanisms to prevent them from being repeatedly sampled. Fluid Benchmarking (Hofmann et al., 2025) demonstrates the consequence: under random sampling, a mislabeled item appears in nearly every evaluation, whereas adaptive IRT-based selection surfaces one only after roughly 100 sessions. This illustrates how adaptive, information-based item selection can mitigate

the impact of low-quality items. These findings collectively reinforce the need for psychometric validation in future benchmark design, including procedures such as discrimination screening and content-alignment checks.

Second, **model fit** is essential for trustworthy item parameter estimation, yet it is rarely examined or reported in existing benchmark-reduction methods that apply IRT. When the underlying IRT model fits poorly, the resulting difficulty and discrimination estimates become unstable, which in turn compromises any downstream conclusions drawn from reduced item sets. Our results show that several widely used static subsets exhibit substantial misfit (Table 1), underscoring that model-fit diagnostics should be a standard requirement for any benchmark that applies IRT for item calibration or reduction. Routine reporting enables researchers to verify whether the assumed psychometric model adequately captures LLM response behavior before relying on the resulting item parameters or reduced test forms.

Third, when item banks are partitioned for computational feasibility, **linking** procedures become crucial. Partitioning items without proper linking can introduce scale drift if parameters are estimated independently across subsets. Common-person linking, where the same set of models responds to all partitions, ensures that items are placed on a consistent latent scale. This preserves the interpretability of difficulty and discrimination estimates and supports coherent benchmarking even when calibration must be distributed or performed in stages. Future large-scale benchmarks should adopt principled linking strategies to maintain comparability across domains and bank updates.

Finally, reduced item sets necessitate **content balancing** to preserve assessment validity. Content balancing ensures proportional representation across skill domains or cognitive competencies, preventing benchmarks from overemphasizing specific subskills. Without it, evaluations risk becoming biased or unrepresentative. Our adaptive framework can be readily extended to jointly optimize domain coverage and measurement precision (Cheng & Chang, 2009). Achieving comparable balance in static reduced subsets is far more difficult and typically requires extensive manual tuning or domain expertise.

### 5.2 LIMITATIONS AND FUTURE WORK

Despite notable efficiency gains, our framework has several limitations. Initial calibration relies on a representative model population, which may become outdated as architectures evolve. Nonetheless, adaptive testing supports incremental updates: new model responses can be incorporated to refine item parameters and maintain calibration accuracy over time. The current implementation remains limited to multiple-choice formats, while generative or open-ended tasks require alternative scoring and modeling strategies. Moreover, our framework is unidimensional, assuming a single latent proficiency dimension across all items, whereas LLM capabilities are inherently multidimensional. Future work should extend the framework to multidimensional IRT formulations that jointly model reasoning, factuality, and linguistic ability.

To facilitate broader adoption, we have implemented a modular scoring interface that allows users to define custom evaluation functions (e.g., determining whether a model answers a selected item correctly). Future work will explore online or Bayesian calibration techniques for continuous item updating, multidimensional modeling to capture diverse LLM capabilities, and hybrid adaptive–generative evaluation for open-ended tasks. We also plan to extend the system with interfaces for cross-model benchmarking, enhanced item exposure control, and visualization tools that improve interpretability and diagnostic insight.

### 6 CONCLUSION

We presented ATLAS, a large-scale adaptive testing framework that reframes LLM evaluation by moving beyond fixed-form, accuracy-based benchmarking toward dynamic ability estimation. Through psychometric modeling and information-guided item selection, ATLAS achieves up to 90% item reduction, avoids accuracy ceiling effects, and reveals ability differences that static benchmarks overlook. Our analysis further highlights the importance of rigorous model-fit validation, item-quality assessment, and principled linking procedures for building reliable and scalable benchmarks. These advances show that adaptive, psychometrically grounded evaluation offers a more efficient, interpretable, and robust foundation for assessing the rapidly growing landscape of LLMs.

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

## A  DETAILED ANALYSIS OF AVERAGE SCORE LIMITATIONS

Average score (percent correct) remains the most widely reported metric for evaluating LLMs. While it provides a convenient ordinal indicator for fixed forms, it is a shaky measure of underlying ability.

First, average scores are *form-dependent*: changing the mix or difficulty of items alters percent correct, even if the model's true ability is unchanged. Second, the metric has a *nonlinear scale*: improvements at the extremes (e.g., $98\% \rightarrow 100\%$) do not reflect the same underlying gain as improvements in the middle (e.g., $50\% \rightarrow 52\%$). Third, it assumes *equal informativeness* across items, allowing easy or guessable items to influence the mean as much as highly discriminative ones. Fourth, it is subject to *coverage bias*: the observed score reflects the content blueprint of the test rather than ability across domains. Fifth, average scores offer *no measure of uncertainty*, making it unclear whether differences are statistically meaningful. Finally, they are highly *sensitive to contamination*: memorized items from pretraining can artificially inflate percent correct without reflecting genuine reasoning or generalization.

In contrast, IRT-based ability estimates ($\theta$) provide form-invariant, uncertainty-aware measures that adjust for item difficulty and discrimination. Reporting $\theta \pm \mathrm{SE}(\theta)$ offers a psychometrically principled alternative. For communication purposes, reconstructed percent scores may be shown alongside, but $\theta$ should serve as the primary indicator of model capability.

## B  COMPARISON OF IRT-BASED BENCHMARK METHODS

Table 4: Comparison of IRT-Based Benchmark Methods for LLMs

| **Factor** | **TinyBenchmarks (Static)** | **MetaBench (Static)** | **ATLAS** |
|---|---|---|---|
| IRT Calibration | Required | Required | Required |
| Adaptivity | None (same items) | None (same items) | High (items vary by ability) |
| Test Length | Fixed | Fixed | Variable, stopping rules |
| Exposure control | High (same items reused) | High (same items reused) | Low (rotating pool) |
| Pool Sensitivity | Subset dependent | Subset dependent | Robust to large pools |
| Fairness | Biased if mis-targeted | Biased if mis-targeted | Balanced across abilities |
| Score Precision | Low at extremes | Low at extremes | High, SEs available |
| Model Fit | Rarely checked | Rarely checked | Possible fit checks |
| Saturation Risk | High | High | Low |

## C  DATA PREPROCESSING DETAILS

### C.1  POINT-BISERIAL CORRELATION FORMULA

The point-biserial correlation (Allen & Yen, 2001) for item $i$ is defined as:

$$r_{pb}(i) \;=\; \frac{\bar{T}_{\ell|Y_{i\ell}=1} - \bar{T}_{\ell|Y_{i\ell}=0}}{s_T} \cdot \sqrt{p_i q_i},$$

where $\bar{T}_{\ell|Y_{i\ell}=1}$ and $\bar{T}_{\ell|Y_{i\ell}=0}$ are the mean total scores of models that answered item $i$ correctly and incorrectly, respectively; $s_T$ is the standard deviation of total scores $\{T_\ell\}$; $p_i = \frac{1}{|\mathcal{L}|}\sum_{\ell \in \mathcal{L}} Y_{i\ell}$ is the proportion of models that answered item $i$ correctly; $q_i = 1 - p_i$; and $|\mathcal{L}|$ is the number of models.

## C.2 Detailed Explanation of the Common-Person Calibration Procedure

In this work, calibration refers to estimating item characteristics under the 3PL item response theory model. The purpose of calibration is to place all items on a shared difficulty scale so that performance comparisons across items and models become meaningful. Under the 3PL model, each item is described by a difficulty parameter, a discrimination parameter that captures how strongly the item differentiates between high- and low-performing models, and a guessing parameter that reflects the chance of a correct response when the model effectively guesses. When every LLM responds to the same items, their collective performance patterns allow these parameters to be estimated in a consistent way. This provides a principled way to identify which items are easy or difficult for models and which items are more or less informative. Calibrating the full benchmark at once would be computationally intensive because the 3PL model becomes more expensive to fit as the number of items grows. To make the process tractable, we divide the full item pool into several non-overlapping subsets and calibrate each subset independently. This reduces the computational load substantially, but it also means that each subset is estimated on its own internal scale. For example, the notion of "difficulty" in one subset is not automatically aligned with the notion of "difficulty" in another. A separate linking step is therefore required to place all subsets onto a shared scale.

Linking requires shared reference points known as anchors. In educational measurement, anchors are typically common items or common examinees that appear in multiple test forms. They serve as a bridge that allows independently calibrated scales to be aligned. In our setting, every LLM responds to every item in the benchmark, which means that the same population of models appears in the calibration of each item subset. The models therefore act as common persons in the traditional psychometric sense and serve as the linking anchors for the benchmark. Their relative performance across subsets provides the information needed to align the scale of each subset with the others. The linking procedure examines how the same models perform across the different subsets and adjusts each subset's scale so that the overall performance patterns match. If a model appears stronger than its peers in one subset and shows a similar relative standing in another, then the two subsets can be placed on the same scale by aligning the average performance level and the overall spread of performance. This rescaling is then applied to the item parameters of each subset so that all items, regardless of which subset they came from, are expressed on a single, unified difficulty metric.

This form of common-person linking is particularly effective for LLM benchmarking. In human testing, it is rarely feasible for every examinee to respond to every item, and linking must rely on smaller or less reliable anchor sets. LLMs do not face constraints such as fatigue, practice effects, or time limits, which allows us to use the entire model population as a complete and stable set of anchors. This makes the linking process highly robust and enables a scalable calibration framework that achieves substantial computational efficiency while maintaining coherence across a very large item bank.

## C.3 Calibration Data, Item-Bank Partitioning, and Fit Statistics

Table 5: Statistics describing the calibration dataset, testing dataset, item-bank size after filtering, number of calibration partitions (K), and average model–data fit (RMSEA from M2) across all benchmarks.

|  | WinoGrande | TruthfulQA | HellaSwag | GSM8K | ARC |
|---|---|---|---|---|---|
| **Models used for calibration** | 4680 | 4635 | 3467 | 3775 | 3747 |
| **Models used for testing** | 521 | 516 | 386 | 420 | 417 |
| **Calibration subsets ($K$)** | 10 | 6 | 50 | 12 | 8 |
| **Items after filtering** | 1045 | 627 | 5600 | 1306 | 839 |
| **Average RMSEA ($M_2$)** | 0.0565 | 0.0690 | 0.0482 | 0.0438 | 0.0595 |

# D Detailed Experimental Setup and Metrics

## D.1 Benchmarks and Datasets

We conduct experiments on five diverse benchmarks covering different cognitive domains:

- **WinoGrande**: Commonsense reasoning with pronoun resolution
- **TruthfulQA**: Factual consistency and truthfulness evaluation
- **HellaSwag**: Procedural inference and common sense completion
- **GSM8K**: Mathematical word problems requiring multi-step reasoning
- **ARC**: Scientific question answering across multiple domains

All experiments use the calibrated item banks from Section 3.3, ensuring consistent filtering and parameter quality across datasets.

## D.2 Baseline Configurations

We compare ATLAS against four fixed, non-adaptive strategies:

**Random Baseline:** Samples 100 items uniformly from the full bank without consideration of item parameters or model ability.

**TinyBenchmarks:** Uses the predetermined subset from Polo et al. (2024), selected via clustering methods but without explicit Fisher information optimization for ability estimation.

**MetaBench-Primary and MetaBench-Secondary:** Curated splits from Kipnis et al. (2025) that require computationally expensive iterations to identify stable subsets. These splits emphasize predictive accuracy over psychometric validity.

All baseline data is available on Hugging Face: tinyBenchmarks, HCAI/metabench.

Unlike ATLAS, these approaches do not adapt to individual test-takers and serve only as static reference points for accuracy–efficiency tradeoffs.

## D.3 ATLAS Configuration Details

For each model $\ell$, we run ATLAS under three precision-based stopping thresholds:

- $\text{SE}(\hat{\theta}) \leq 0.1$: High precision, suitable for fine-grained model comparison
- $\text{SE}(\hat{\theta}) \leq 0.2$: Moderate precision, balancing accuracy and efficiency
- $\text{SE}(\hat{\theta}) \leq 0.3$: Lower precision, maximizing efficiency for rapid screening

A minimum of 30 items is enforced to prevent premature termination due to lucky guesses or initial high-information items, while the maximum is capped at 500 items to ensure computational feasibility. This setup balances precision and budget constraints, simulating realistic conditions for adaptive evaluation in production environments.

# E   EVALUATION METRIC DEFINITIONS

This section provides the exact mathematical definitions of the evaluation metrics introduced in Section 4, along with brief interpretations.

**Average Mean Absolute Error (MAE) and Standard Error (SE).**   We compute MAE for both ability estimates and accuracy scores. For ability, let $\hat{\theta}_\ell$ denote the CAT-derived estimate and $\hat{\theta}_\ell^{\text{whole}}$ the full-bank reference. The ability MAE is

$$\text{MAE}_\theta = \frac{1}{|\mathcal{L}|} \sum_{\ell \in \mathcal{L}} \left| \hat{\theta}_\ell - \hat{\theta}_\ell^{\text{whole}} \right|.$$

For accuracy, let $\widehat{Acc}_\ell^{\text{p-IRT}}$ denote the reconstructed accuracy (e.g., via p-IRT) and $\text{Acc}_\ell^{\text{raw}}$ the observed raw accuracy. The accuracy MAE is

$$\text{MAE}_{\text{acc}} = \frac{1}{|\mathcal{L}|} \sum_{\ell \in \mathcal{L}} \left| \widehat{Acc}_\ell^{\text{p-IRT}} - \text{Acc}_\ell^{\text{raw}} \right|.$$

To quantify variability across models, we also report the standard error (SE) of MAE. Let $e_\ell$ denote the per-model absolute error and $\overline{e}$ its mean. The standard deviation (SD) is

$$\text{SD} = \sqrt{\frac{1}{|\mathcal{L}| - 1} \sum_{\ell \in \mathcal{L}} (e_\ell - \overline{e})^2},$$

and the standard error (SE) is

$$\text{SE} = \frac{\text{SD}}{\sqrt{|\mathcal{L}|}}.$$

*Interpretation:* Lower MAE and SE indicate higher fidelity and greater stability: CAT-derived estimates more closely match whole-bank references (for ability) or observed scores (for accuracy) and do so consistently across models.

**Information Efficiency Score (IES).**   To compare the efficiency of different evaluation methods, we define the *Information Efficiency Score* (IES) relative to a baseline of 100-item uniform random sampling (Random_100). For a given method, let $\text{MAE}_{\text{method}}$ denote its average MAE and $\text{MAE}_{\text{Random}}$ the MAE under Random_100. Let $\text{Items}_{\text{method}}$ denote the average number of selected subset items. The IES is:

$$\text{IES} \; = \; \left( \frac{\text{MAE}_{\text{method}}}{\text{MAE}_{\text{Random}}} \right) \left( \frac{\text{Items}_{\text{method}}}{100} \right). \tag{4}$$

*Interpretation:* An IES value below 1 indicates that the method achieves a better accuracy–efficiency tradeoff than the Random_100 baseline, requiring fewer items and/or producing lower error for the same number of items. An IES value of 1 means the method is equally efficient as Random_100. Values greater than 1 indicate lower efficiency, meaning the method uses more items and/or yields higher error than the baseline.

**Average Item Exposure Rate.**   Let $h_i$ denote the number of models administered item $i$, with $|\mathcal{I}|$ total items and $|\mathcal{L}|$ total models. The item exposure probability for item $i$ is

$$P(A_i) = \frac{h_i}{|\mathcal{L}|}. \tag{5}$$

The average item exposure rate is then

$$\bar{P}(A_i) = \frac{1}{|\mathcal{I}|} \sum_{i \in \mathcal{I}} P(A_i). \tag{6}$$

*Interpretation:* Lower values indicate higher adaptivity and greater item diversity, while higher values suggest uniform or repetitive item usage across models.

**Test Overlap Rate.** Following Chen (2005), the expected proportion of common items between two randomly selected test forms is given by

$$\bar{Q} = \frac{|\mathcal{L}| \sum_{i=1}^{|\mathcal{I}|} P(A_i)^2}{\bar{L}(|\mathcal{L}| - 1)} - \frac{1}{|\mathcal{L}| - 1}, \tag{7}$$

where $\bar{L}$ is the average test length.

*Interpretation:* Lower values of $\bar{Q}$ imply greater test form diversity, which reduces risks of collusion and item memorization.

## E.1 Correlation Metrics

For completeness, we provide the definitions of the rank-based correlation coefficients used in Section 4.

**Spearman correlation.**

$$\rho = 1 - \frac{6 \sum_{i=1}^{n} d_i^2}{n(n^2 - 1)},$$

where $d_i$ is the rank difference for observation $i$ across the two measures.

**Kendall correlation.**

$$\tau = \frac{(\#\text{concordant pairs}) - (\#\text{discordant pairs})}{\frac{1}{2} n(n-1)}.$$

## F  Performance-IRT (p-IRT) Estimator

The Performance-IRT (p-IRT) estimator (Polo et al., 2024) is a probabilistic scoring method used to compute expected accuracy when only a subset of benchmark items is observed. Conceptually, p-IRT is grounded in the Test Characteristic Curve (TCC) (Lord & Novick, 2008; Hambleton et al., 1991), which maps a model's ability $\hat{\theta}$ to its expected probability of correctly answering items under the calibrated 3PL model. It provides an estimate of a model's overall benchmark accuracy without requiring evaluation on the full item set.

**Goal.** We formulate LLM evaluation as a psychometric measurement problem. Let $\mathcal{I}$ denote the full set of benchmark items and $\mathcal{L}$ the set of language models. For each model $\ell \in \mathcal{L}$ and item $i \in \mathcal{I}$, we observe a binary response $Y_{i,\ell} \in \{0, 1\}$, forming the item–response matrix $\{Y_{i,\ell}\}_{i \in \mathcal{I}, \ell \in \mathcal{L}}$. The true full-benchmark accuracy of model $\ell$ is

$$\text{Acc}_\ell^{\text{raw}} = \frac{1}{|\mathcal{I}|} \sum_{i \in \mathcal{I}} Y_{i,\ell}.$$

The p-IRT estimator approximates $\text{Acc}_\ell^{\text{raw}}$ when only a subset of items $\widehat{\mathcal{I}} \subseteq \mathcal{I}$ is observed, by combining the model's observed responses on $\widehat{\mathcal{I}}$ with IRT-predicted probabilities on the remaining items $\mathcal{I} \setminus \widehat{\mathcal{I}}$.

**Estimator.** P-IRT computes the conditional expectation

$$\widehat{Acc}_\ell^{\text{p-IRT}} = \mathbb{E}\left[ \text{Acc}_\ell^{\text{raw}} \,\middle|\, \{Y_{i,\ell} : i \in \widehat{\mathcal{I}}\} \right],$$

which is the minimum–mean-squared-error predictor of $\text{Acc}_\ell^{\text{raw}}$ under the calibrated IRT model. Then

$$\widehat{Acc}_\ell^{\text{p-IRT}} = \frac{|\widehat{\mathcal{I}}|}{|\mathcal{I}|} \cdot \underbrace{\frac{1}{|\widehat{\mathcal{I}}|} \sum_{i \in \widehat{\mathcal{I}}} Y_{i,\ell}}_{\text{Observed accuracy}} + \frac{|\mathcal{I} \setminus \widehat{\mathcal{I}}|}{|\mathcal{I}|} \cdot \underbrace{\frac{1}{|\mathcal{I} \setminus \widehat{\mathcal{I}}|} \sum_{i \in \mathcal{I} \setminus \widehat{\mathcal{I}}} \hat{p}_{i,\ell}}_{\text{Unobserved TCC}},$$

where

$$\hat{p}_{i,\ell} = P\left( Y_{i,\ell} = 1 \mid \hat{\theta}_\ell, \hat{a}_i, \hat{b}_i, \hat{c}_i \right)$$

is the predicted probability of correctness for model $\ell$ under the calibrated 3PL model.

**Intuition.** The p-IRT estimator is a weighted combination of:

- **Observed accuracy** on the subset $\widehat{\mathcal{I}}$.
- **Unobserved TCC** on the remaining items $\mathcal{I} \setminus \widehat{\mathcal{I}}$, based on the model's ability $\hat{\theta}_\ell$ and item parameters.

The weight $|\widehat{\mathcal{I}}|/|\mathcal{I}| \in [0, 1]$ corresponds to the proportion of observed items and determines the tradeoff between observed and predicted performance.

**Use in This Work.** We apply p-IRT to reconstruct accuracy from ability estimates $\hat{\theta}_\ell$. As shown in Table 3, reconstructed accuracies closely match raw accuracies across benchmarks, confirming that ability estimates retain the global performance structure while smoothing noise and offering finer discrimination than raw accuracy alone.

# G ADDITIONAL EXPERIMENTAL RESULTS

## G.1 INTERPRETING TEST OVERLAP, EXPOSURE, AND RUNTIME IN ATLAS

Table 6 provides a detailed breakdown of adaptive evaluation behavior across all five benchmarks, summarizing test overlap, average item exposure, and selection time. These metrics together illustrate how ATLAS balances efficiency, diversity, and computational scalability when administering adaptive tests. Formal definitions of all metrics are included in Appendix E.

Test overlap rates quantify how frequently different models are exposed to the same items. Across benchmarks, overlap remains modest, ranging from roughly 11% to 23%. These values are far lower than those of static subsets, which administer identical items to all models. The relatively low overlap indicates that ATLAS tailors item sequences to each model's evolving ability estimate rather than relying on a fixed set of questions. For example, HellaSwag reaches the lowest overlap values (as low as 11.26%), reflecting its large item pool and the wide range of informative items available. Higher overlap on datasets such as GSM8K (approximately 20–24%) reflects the smaller bank size and the concentration of discriminative items in particular ability regions. Overall, the overlap statistics confirm that ATLAS provides genuine adaptivity while preserving comparability across models.

Average item exposure rates remain low across all settings, consistently under 12% and often much lower. Exposure values around 3–5% on HellaSwag and WinoGrande (for SE thresholds of 0.2 and 0.3) indicate that ATLAS does not rely excessively on a small subset of items. Low exposure reduces the risk of memorization or contamination in long-term evaluation scenarios, broadens the portion of the item bank that contributes to measurement, and ensures that no individual item disproportionately influences ability estimation. The pattern across SE thresholds reflects a standard property of adaptive testing: when fewer items are required (larger SE thresholds), exposure becomes more concentrated on the most informative items. In ATLAS, this concentration remains moderate, indicating healthy rotation among informative items.

Selection time reflects the computational cost of the full adaptive selection loop, including Fisher information computation and termination checks. It refers to the complete runtime of the adaptive item selection loop for each model, rather than the time required for a single item decision. Times range from 9 to 76 seconds per model and scale predictably with benchmark size. TruthfulQA, with 628 items, achieves the fastest selection times (approximately 9–16 seconds). In contrast, HellaSwag, with more than 5600 items, shows the longest selection times (57–76 seconds), due to the larger number of items evaluated when determining the most informative question at each step. Importantly, even in the largest setting, selection remains well under 90 seconds, and for all other benchmarks it typically completes within tens of seconds. This confirms that ATLAS is computationally practical for both interactive evaluation and large-scale benchmarking workflows.

Taken together, the results in Table 6 show that ATLAS achieves high adaptivity, broad item utilization, and practical runtime efficiency across diverse benchmarks. Low overlap and exposure promote content coverage and robustness, while stable runtime performance ensures operational scalability without compromising statistical quality.

Table 6: Adaptive evaluation efficiency and diversity. ATLAS maintains low item exposure rates ($< 12\%$) and moderate test overlap ($13 - 24\%$) with fast selection times ($< 76$ seconds per model). Lower values indicate better performance for all metrics.

| Benchmark (Item #) | Method | Test Overlap ↓ Rate (%) | Avg. Item ↓ Exposure Rate (%) | Avg. Selection ↓ Time (s) |
|---|---|---|---|---|
| WinoGrande (1046) | ATLAS$_{SE \leq 0.1}$ | 18.22 | 8.24 | 40.99 |
| | ATLAS$_{SE \leq 0.2}$ | **14.93** | 4.71 | 19.92 |
| | ATLAS$_{SE \leq 0.3}$ | 17.03 | **4.04** | **16.74** |
| TruthfulQA (628) | ATLAS$_{SE \leq 0.1}$ | **17.32** | **7.86** | 15.97 |
| | ATLAS$_{SE \leq 0.2}$ | 18.43 | 9.58 | **9.37** |
| | ATLAS$_{SE \leq 0.3}$ | 18.07 | 9.49 | 9.72 |
| HellaSwag (5608) | ATLAS$_{SE \leq 0.1}$ | **11.26** | **3.86** | 75.52 |
| | ATLAS$_{SE \leq 0.2}$ | 13.72 | 4.78 | **56.93** |
| | ATLAS$_{SE \leq 0.3}$ | 13.67 | 4.82 | 57.06 |
| GSM8K (1307) | ATLAS$_{SE \leq 0.1}$ | 21.27 | 7.21 | 45.69 |
| | ATLAS$_{SE \leq 0.2}$ | **20.78** | **4.40** | 24.08 |
| | ATLAS$_{SE \leq 0.3}$ | 23.70 | 5.54 | **19.06** |
| ARC (842) | ATLAS$_{SE \leq 0.1}$ | 19.15 | 11.15 | 30.99 |
| | ATLAS$_{SE \leq 0.2}$ | **17.09** | **5.41** | 13.98 |
| | ATLAS$_{SE \leq 0.3}$ | 19.60 | 9.21 | **11.82** |

## G.2 EXPERIMENT ON MMLU

While both *TinyBenchmarks* (Polo et al., 2024) and *MetaBench* (Kipnis et al., 2025) include MMLU (Hendrycks et al., 2020) as part of their evaluation suites, they treat it as a single unified dataset by aggregating all 57 subject areas. In contrast, we perform evaluation on a per-subject basis. This design choice acknowledges the heterogeneous nature of MMLU, where each subject represents a distinct knowledge domain with varying linguistic characteristics, content distributions, and difficulty levels. Aggregating across all subjects can obscure these domain-specific patterns and limit interpretability in adaptive assessment.

The corresponding results are reported in Table 7. Despite the small number of items per subject, ATLAS consistently demonstrates robust adaptive evaluation performance. As the selection threshold is relaxed ($SE \leq 0.1 \rightarrow 0.3$), the mean absolute error (MAE) increases moderately (e.g., from 0.099 to 0.235 in *Anatomy*), while the number of evaluated items is substantially reduced (approximately 80%). This indicates that ATLAS effectively balances efficiency and accuracy, even in limited-data regimes.

Moreover, reductions in test overlap and exposure rates across the evaluated subjects suggest that the adaptive mechanism achieves broader item coverage and mitigates redundancy. Evaluation time

also decreases proportionally with the number of items, confirming the computational efficiency of the adaptive process.

| Benchmark | Method | MAE ↓ | Avg. Item ↓ | Test Overlap ↓ | Exposure Rate ↓ | Avg. Time (s) ↓ |
|---|---|---|---|---|---|---|
| MMLU-Abstract Algebra (79 items) | ATLAS$_{SE \leq 0.1}$ | **0.025** | 52.45 | 0.6808 | 0.6639 | 0.59 |
| | ATLAS$_{SE \leq 0.2}$ | 0.067 | 32.18 | 0.4413 | 0.4073 | 0.36 |
| | ATLAS$_{SE \leq 0.3}$ | 0.098 | **19.90** | **0.3169** | **0.2519** | **0.22** |
| MMLU-Anatomy (113 items) | ATLAS$_{SE \leq 0.1}$ | **0.099** | 100.00 | 0.94 | 0.88 | 4.93 |
| | ATLAS$_{SE \leq 0.2}$ | 0.149 | 53.29 | 0.54 | 0.47 | 2.70 |
| | ATLAS$_{SE \leq 0.3}$ | 0.235 | **20.49** | **0.32** | **0.18** | **0.98** |
| MMLU-Astronomy (136 items) | ATLAS$_{SE \leq 0.1}$ | **0.099** | 93.22 | 0.82 | 0.69 | 6.30 |
| | ATLAS$_{SE \leq 0.2}$ | 0.157 | 48.21 | 0.48 | 0.35 | 3.29 |
| | ATLAS$_{SE \leq 0.3}$ | 0.235 | **11.80** | **0.40** | **0.11** | **0.79** |
| MMLU-Business Ethics (95 items) | ATLAS$_{SE \leq 0.1}$ | **0.040** | 86.61 | 0.9122 | 0.9117 | 4.09 |
| | ATLAS$_{SE \leq 0.2}$ | 0.072 | 54.43 | 0.5912 | 0.5729 | 5.29 |
| | ATLAS$_{SE \leq 0.3}$ | 0.160 | **13.46** | **0.4361** | **0.1417** | **0.63** |
| MMLU-Clinical Knowledge (198 items) | ATLAS$_{SE \leq 0.1}$ | **0.044** | 99.86 | 0.7302 | 0.5043 | 8.69 |
| | ATLAS$_{SE \leq 0.2}$ | 0.186 | 23.33 | 0.2894 | 0.1477 | 4.20 |
| | ATLAS$_{SE \leq 0.3}$ | 0.244 | **10.46** | **0.2537** | **0.1016** | **0.91** |
| MMLU-College Biology (131 items) | ATLAS$_{SE \leq 0.1}$ | **0.031** | 100.00 | 0.8870 | 0.7634 | 5.98 |
| | ATLAS$_{SE \leq 0.2}$ | 0.083 | 67.87 | 0.6168 | 0.5186 | 4.20 |
| | ATLAS$_{SE \leq 0.3}$ | 0.134 | **29.40** | **0.3380** | **0.2315** | **1.90** |
| MMLU-College Chemistry (77 items) | ATLAS$_{SE \leq 0.1}$ | **0.029** | 77.00 | 1.0000 | 1.0000 | 0.72 |
| | ATLAS$_{SE \leq 0.2}$ | 0.070 | 52.23 | 0.6857 | 0.6783 | 0.47 |
| | ATLAS$_{SE \leq 0.3}$ | 0.116 | **17.84** | **0.4230** | **0.2308** | **0.25** |
| MMLU-College Computer Science (84 items) | ATLAS$_{SE \leq 0.1}$ | **0.030** | 82.71 | 0.9845 | 0.9844 | 0.73 |
| | ATLAS$_{SE \leq 0.2}$ | 0.066 | 41.67 | 0.5305 | 0.4961 | 0.39 |
| | ATLAS$_{SE \leq 0.3}$ | 0.094 | **13.59** | **0.3367** | **0.1621** | **0.19** |
| MMLU-College Mathematics (69 items) | ATLAS$_{SE \leq 0.1}$ | **0.025** | 63.86 | 0.9267 | 0.9256 | 0.62 |
| | ATLAS$_{SE \leq 0.2}$ | 0.062 | 44.67 | 0.6669 | 0.6478 | 0.43 |
| | ATLAS$_{SE \leq 0.3}$ | 0.089 | **28.48** | **0.4832** | **0.4128** | **0.29** |
| MMLU-College Medicine (157 items) | ATLAS$_{SE \leq 0.1}$ | **0.038** | 100.00 | 0.8039 | 0.6369 | 7.02 |
| | ATLAS$_{SE \leq 0.2}$ | 0.111 | 66.64 | 0.5986 | 0.4350 | 4.90 |
| | ATLAS$_{SE \leq 0.3}$ | 0.171 | **27.77** | **0.3062** | **0.2061** | **2.02** |
| MMLU-College Physics (72 items) | ATLAS$_{SE \leq 0.1}$ | **0.020** | 69.39 | 0.9634 | 0.9632 | 0.69 |
| | ATLAS$_{SE \leq 0.2}$ | 0.041 | 61.78 | 0.8618 | 0.8580 | 0.63 |
| | ATLAS$_{SE \leq 0.3}$ | 0.074 | **30.69** | **0.5167** | **0.4264** | **0.36** |
| MMLU-Computer Security (84 items) | ATLAS$_{SE \leq 0.1}$ | **0.019** | 84.00 | 1.0000 | 1.0000 | 0.71 |
| | ATLAS$_{SE \leq 0.2}$ | 0.034 | 75.99 | 0.9067 | 0.9049 | 0.65 |
| | ATLAS$_{SE \leq 0.3}$ | 0.065 | **29.54** | **0.4484** | **0.3508** | **0.35** |
| MMLU-Conceptual Physics (203 items) | ATLAS$_{SE \leq 0.1}$ | **0.042** | 69.43 | 0.5217 | 0.3417 | 4.02 |
| | ATLAS$_{SE \leq 0.2}$ | 0.160 | 13.86 | 0.1844 | 0.0713 | 0.98 |
| | ATLAS$_{SE \leq 0.3}$ | 0.205 | **10.58** | **0.1844** | **0.0596** | **0.78** |
| MMLU-Econometrics (102 items) | ATLAS$_{SE \leq 0.1}$ | **0.031** | 91.67 | 0.9026 | 0.8992 | 5.38 |
| | ATLAS$_{SE \leq 0.2}$ | 0.074 | 56.23 | 0.5678 | 0.5508 | 4.05 |
| | ATLAS$_{SE \leq 0.3}$ | 0.148 | **13.44** | **0.4002** | **0.1442** | **0.64** |
| MMLU-Electrical Engineering (126 items) | ATLAS$_{SE \leq 0.1}$ | **0.032** | 100.00 | 0.8917 | 0.7937 | 5.98 |
| | ATLAS$_{SE \leq 0.2}$ | 0.081 | 61.94 | 0.5710 | 0.4917 | 4.05 |
| | ATLAS$_{SE \leq 0.3}$ | 0.134 | **18.70** | **0.3021** | **0.1498** | **1.05** |
| MMLU-Elementary Mathematics (220 items) | ATLAS$_{SE \leq 0.1}$ | **0.050** | 66.10 | 0.4546 | 0.3006 | 3.80 |
| | ATLAS$_{SE \leq 0.2}$ | 0.191 | 13.34 | 0.2313 | 0.0843 | 0.87 |
| | ATLAS$_{SE \leq 0.3}$ | 0.237 | **10.02** | **0.2020** | **0.0652** | **0.67** |
| MMLU-Formal Logic (109 items) | ATLAS$_{SE \leq 0.1}$ | **0.030** | 91.05 | 0.8751 | 0.8359 | 5.43 |
| | ATLAS$_{SE \leq 0.2}$ | 0.098 | 31.64 | 0.3807 | 0.2895 | 2.18 |
| | ATLAS$_{SE \leq 0.3}$ | 0.146 | **20.89** | **0.2762** | **0.1919** | **1.26** |
| MMLU-Global Facts (81 items) | ATLAS$_{SE \leq 0.1}$ | **0.022** | 65.09 | 0.8060 | 0.8041 | 0.56 |
| | ATLAS$_{SE \leq 0.2}$ | 0.038 | 57.62 | 0.7158 | 0.7115 | 0.50 |
| | ATLAS$_{SE \leq 0.3}$ | 0.069 | **18.03** | **0.4382** | **0.2225** | **0.22** |
| MMLU-High School Biology (251 items) | ATLAS$_{SE \leq 0.1}$ | **0.037** | 100.00 | 0.6336 | 0.3984 | 6.09 |
| | ATLAS$_{SE \leq 0.2}$ | 0.124 | 37.74 | 0.2869 | 0.1932 | 2.78 |
| | ATLAS$_{SE \leq 0.3}$ | 0.171 | **25.49** | **0.2193** | **0.1480** | **1.98** |
| MMLU-High School Chemistry (169 items) | ATLAS$_{SE \leq 0.1}$ | **0.033** | 100.00 | 0.7729 | 0.5917 | 5.64 |
| | ATLAS$_{SE \leq 0.2}$ | 0.111 | 36.56 | 0.3257 | 0.2349 | 2.62 |
| | ATLAS$_{SE \leq 0.3}$ | 0.151 | **17.30** | **0.2078** | **0.1182** | **1.00** |

*Continued on next page*

| Benchmark | Method | MAE ↓ | Avg. Item ↓ | Test Overlap ↓ | Exposure Rate ↓ | Avg. Time (s) ↓ |
|---|---|---|---|---|---|---|
| MMLU-High School Computer Science (94 items) | ATLAS$_{SE \leq 0.1}$ | **0.021** | 94.00 | 1.0000 | 1.0000 | 0.74 |
| | ATLAS$_{SE \leq 0.2}$ | 0.041 | 73.23 | 0.7911 | 0.7788 | 0.59 |
| | ATLAS$_{SE \leq 0.3}$ | 0.066 | **31.34** | **0.4380** | **0.3330** | **0.34** |
| MMLU-High School European History (147 items) | ATLAS$_{SE \leq 0.1}$ | **0.038** | 100.00 | 0.8630 | 0.6803 | 6.31 |
| | ATLAS$_{SE \leq 0.2}$ | 0.081 | 75.10 | 0.6755 | 0.5368 | 5.16 |
| | ATLAS$_{SE \leq 0.3}$ | 0.137 | **27.05** | **0.3238** | **0.2001** | **2.15** |
| MMLU-High School Geography (170 items) | ATLAS$_{SE \leq 0.1}$ | **0.036** | 85.66 | 0.6861 | 0.5039 | 4.89 |
| | ATLAS$_{SE \leq 0.2}$ | 0.095 | 45.02 | 0.4002 | 0.2904 | 2.90 |
| | ATLAS$_{SE \leq 0.3}$ | 0.142 | **17.80** | **0.3141** | **0.1356** | **1.19** |
| MMLU-High School Government & Politics (154 items) | ATLAS$_{SE \leq 0.1}$ | **0.033** | 100.00 | 0.8509 | 0.6494 | 5.94 |
| | ATLAS$_{SE \leq 0.2}$ | 0.062 | 92.63 | 0.7887 | 0.6011 | 5.54 |
| | ATLAS$_{SE \leq 0.3}$ | 0.124 | **26.66** | **0.3425** | **0.1989** | **1.69** |
| MMLU-High School Mathematics (199 items) | ATLAS$_{SE \leq 0.1}$ | **0.044** | 56.19 | 0.4137 | 0.2822 | 3.31 |
| | ATLAS$_{SE \leq 0.2}$ | 0.094 | 47.18 | 0.3732 | 0.2486 | 2.86 |
| | ATLAS$_{SE \leq 0.3}$ | 0.183 | **12.12** | **0.2503** | **0.0894** | **0.79** |
| MMLU-High School Microeconomics (193 items) | ATLAS$_{SE \leq 0.1}$ | **0.040** | 93.17 | 0.7021 | 0.4828 | 5.55 |
| | ATLAS$_{SE \leq 0.2}$ | 0.174 | 18.27 | 0.2907 | 0.1188 | 1.25 |
| | ATLAS$_{SE \leq 0.3}$ | 0.223 | **10.64** | **0.2579** | **0.1005** | **0.79** |
| MMLU-High School Macroeconomics (281 items) | ATLAS$_{SE \leq 0.1}$ | **0.089** | 87.94 | 0.48 | 0.31 | 10.49 |
| | ATLAS$_{SE \leq 0.2}$ | 0.197 | 27.64 | 0.22 | 0.13 | 3.46 |
| | ATLAS$_{SE \leq 0.3}$ | 0.240 | **18.26** | **0.20** | **0.09** | **2.34** |
| MMLU-High School Physics (119 items) | ATLAS$_{SE \leq 0.1}$ | **0.026** | 63.30 | 0.5825 | 0.5317 | 3.54 |
| | ATLAS$_{SE \leq 0.2}$ | 0.055 | 51.75 | 0.4900 | 0.4356 | 3.08 |
| | ATLAS$_{SE \leq 0.3}$ | 0.106 | **20.30** | **0.3393** | **0.1932** | **1.37** |
| MMLU-High School Psychology (285 items) | ATLAS$_{SE \leq 0.1}$ | **0.035** | 100.00 | 0.6086 | 0.3509 | 6.19 |
| | ATLAS$_{SE \leq 0.2}$ | 0.147 | 26.80 | 0.2433 | 0.1284 | 1.97 |
| | ATLAS$_{SE \leq 0.3}$ | 0.196 | **11.48** | **0.2101** | **0.0881** | **0.92** |
| MMLU-High School Statistics (185 items) | ATLAS$_{SE \leq 0.1}$ | **0.038** | 69.47 | 0.5587 | 0.3757 | 4.07 |
| | ATLAS$_{SE \leq 0.2}$ | 0.154 | 21.48 | 0.2987 | 0.1219 | 1.67 |
| | ATLAS$_{SE \leq 0.3}$ | 0.205 | **11.81** | **0.2483** | **0.0700** | **1.03** |
| MMLU-High School US History (183 items) | ATLAS$_{SE \leq 0.1}$ | **0.030** | 96.68 | 0.7761 | 0.5283 | 5.77 |
| | ATLAS$_{SE \leq 0.2}$ | 0.069 | 73.31 | 0.6137 | 0.4027 | 4.80 |
| | ATLAS$_{SE \leq 0.3}$ | 0.151 | **13.62** | **0.3348** | **0.1319** | **1.05** |
| MMLU-High School World History (206 items) | ATLAS$_{SE \leq 0.1}$ | **0.031** | 96.43 | 0.7407 | 0.4680 | 5.88 |
| | ATLAS$_{SE \leq 0.2}$ | 0.108 | 40.83 | 0.3806 | 0.2580 | 2.85 |
| | ATLAS$_{SE \leq 0.3}$ | 0.176 | **17.10** | **0.2629** | **0.1230** | **1.28** |
| MMLU-Human Aging (188 items) | ATLAS$_{SE \leq 0.1}$ | **0.034** | 100.00 | 0.7341 | 0.5319 | 5.86 |
| | ATLAS$_{SE \leq 0.2}$ | 0.108 | 40.24 | 0.3546 | 0.2515 | 2.69 |
| | ATLAS$_{SE \leq 0.3}$ | 0.154 | **27.38** | **0.2445** | **0.1802** | **2.00** |
| MMLU-Human Sexuality (116 items) | ATLAS$_{SE \leq 0.1}$ | **0.032** | 100.00 | 0.9137 | 0.8621 | 6.07 |
| | ATLAS$_{SE \leq 0.2}$ | 0.062 | 83.66 | 0.7655 | 0.7212 | 5.25 |
| | ATLAS$_{SE \leq 0.3}$ | 0.154 | **20.43** | **0.2825** | **0.1759** | **1.51** |
| MMLU-International Law (103 items) | ATLAS$_{SE \leq 0.1}$ | **0.028** | 100.00 | 0.9780 | 0.9709 | 5.88 |
| | ATLAS$_{SE \leq 0.2}$ | 0.061 | 76.17 | 0.7582 | 0.7400 | 4.72 |
| | ATLAS$_{SE \leq 0.3}$ | 0.105 | **34.28** | **0.4298** | **0.3333** | **2.57** |
| MMLU-Jurisprudence (99 items) | ATLAS$_{SE \leq 0.1}$ | **0.021** | 99.00 | 1.0000 | 1.0000 | 0.79 |
| | ATLAS$_{SE \leq 0.2}$ | 0.047 | 71.20 | 0.7262 | 0.7191 | 0.60 |
| | ATLAS$_{SE \leq 0.3}$ | 0.095 | **33.57** | **0.3986** | **0.3399** | **0.38** |
| MMLU-Logical Fallacies (147 items) | ATLAS$_{SE \leq 0.1}$ | **0.030** | 100.00 | 0.8562 | 0.6803 | 5.92 |
| | ATLAS$_{SE \leq 0.2}$ | 0.064 | 70.48 | 0.6483 | 0.4928 | 4.73 |
| | ATLAS$_{SE \leq 0.3}$ | 0.117 | **22.21** | **0.2938** | **0.1632** | **1.69** |
| MMLU-Machine Learning (98 items) | ATLAS$_{SE \leq 0.1}$ | **0.027** | 74.29 | 0.7619 | 0.7586 | 3.34 |
| | ATLAS$_{SE \leq 0.2}$ | 0.069 | 39.44 | 0.4402 | 0.4018 | 1.89 |
| | ATLAS$_{SE \leq 0.3}$ | 0.143 | **13.81** | **0.3032** | **0.1410** | **0.71** |
| MMLU-Management (92 items) | ATLAS$_{SE \leq 0.1}$ | **0.022** | 92.00 | 1.0000 | 1.0000 | 0.74 |
| | ATLAS$_{SE \leq 0.2}$ | 0.024 | 90.39 | 0.9831 | 0.9831 | 0.72 |
| | ATLAS$_{SE \leq 0.3}$ | 0.059 | **44.91** | **0.5238** | **0.4881** | **0.47** |
| MMLU-Marketing (178 items) | ATLAS$_{SE \leq 0.1}$ | **0.033** | 100.00 | 0.7974 | 0.5618 | 5.78 |
| | ATLAS$_{SE \leq 0.2}$ | 0.081 | 61.18 | 0.5305 | 0.3645 | 4.06 |
| | ATLAS$_{SE \leq 0.3}$ | 0.134 | **25.09** | **0.3035** | **0.1660** | **1.84** |
| MMLU-Medical Genetics (87 items) | ATLAS$_{SE \leq 0.1}$ | **0.024** | 84.38 | 0.9699 | 0.9698 | 0.74 |
| | ATLAS$_{SE \leq 0.2}$ | 0.045 | 69.31 | 0.7995 | 0.7967 | 0.62 |
| | ATLAS$_{SE \leq 0.3}$ | 0.072 | **55.16** | **0.6449** | **0.6349** | **0.51** |
| MMLU-Miscellaneous (314 items) | ATLAS$_{SE \leq 0.1}$ | **0.048** | 65.51 | 0.3154 | 0.2085 | 3.66 |

| Benchmark | Method | MAE ↓ | Avg. Item ↓ | Test Overlap ↓ | Exposure Rate ↓ | Avg. Time (s) ↓ |
|---|---|---|---|---|---|---|
| | $ATLAS_{SE\leq0.2}$ | 0.156 | 24.00 | 0.1576 | 0.0991 | 1.58 |
| | $ATLAS_{SE\leq0.3}$ | 0.203 | **15.73** | **0.1295** | **0.0697** | **1.10** |
| MMLU-Moral Disputes (229 items) | $ATLAS_{SE\leq0.1}$ | **0.041** | 99.73 | 0.6574 | 0.4355 | 6.03 |
| | $ATLAS_{SE\leq0.2}$ | 0.142 | 27.23 | 0.2498 | 0.1526 | 1.94 |
| | $ATLAS_{SE\leq0.3}$ | 0.195 | **13.45** | **0.2077** | **0.1090** | **0.98** |
| MMLU-Moral Scenarios (486 items) | $ATLAS_{SE\leq0.1}$ | **0.059** | 34.18 | 0.1610 | 0.0704 | 2.24 |
| | $ATLAS_{SE\leq0.2}$ | 0.114 | 19.23 | 0.1216 | 0.0436 | 1.41 |
| | $ATLAS_{SE\leq0.3}$ | 0.158 | **14.14** | **0.0986** | **0.0334** | **1.05** |
| MMLU-Nutrition (244 items) | $ATLAS_{SE\leq0.1}$ | **0.039** | 95.50 | 0.6325 | 0.3914 | 6.00 |
| | $ATLAS_{SE\leq0.2}$ | 0.129 | 38.88 | 0.3028 | 0.1953 | 2.89 |
| | $ATLAS_{SE\leq0.3}$ | 0.176 | **21.25** | **0.2221** | **0.1191** | **1.69** |
| MMLU-Philosophy (214 items) | $ATLAS_{SE\leq0.1}$ | **0.041** | 96.45 | 0.6360 | 0.4510 | 5.90 |
| | $ATLAS_{SE\leq0.2}$ | 0.152 | 23.09 | 0.1983 | 0.1258 | 1.89 |
| | $ATLAS_{SE\leq0.3}$ | 0.202 | **16.56** | **0.1657** | **0.0927** | **1.37** |
| MMLU-Prehistory (265 items) | $ATLAS_{SE\leq0.1}$ | **0.039** | 93.64 | 0.6004 | 0.3531 | 5.72 |
| | $ATLAS_{SE\leq0.2}$ | 0.099 | 44.74 | 0.4126 | 0.2078 | 3.28 |
| | $ATLAS_{SE\leq0.3}$ | 0.190 | **12.29** | **0.2773** | **0.0989** | **0.95** |
| MMLU-Professional Accounting (221 items) | $ATLAS_{SE\leq0.1}$ | **0.047** | 63.18 | 0.4458 | 0.2861 | 3.83 |
| | $ATLAS_{SE\leq0.2}$ | 0.170 | 14.16 | 0.2016 | 0.0682 | 1.03 |
| | $ATLAS_{SE\leq0.3}$ | 0.220 | **10.46** | **0.1899** | **0.0647** | **0.83** |
| MMLU-Professional Law (518 items) | $ATLAS_{SE\leq0.1}$ | **0.064** | 26.94 | 0.1811 | 0.0519 | 1.84 |
| | $ATLAS_{SE\leq0.2}$ | 0.151 | 10.88 | 0.1380 | 0.0283 | 0.98 |
| | $ATLAS_{SE\leq0.3}$ | 0.188 | **10.06** | **0.1308** | **0.0271** | **0.90** |
| MMLU-Professional Medicine (218 items) | $ATLAS_{SE\leq0.1}$ | **0.044** | 95.14 | 0.6919 | 0.4363 | 6.11 |
| | $ATLAS_{SE\leq0.2}$ | 0.167 | 24.91 | 0.3403 | 0.1649 | 2.12 |
| | $ATLAS_{SE\leq0.3}$ | 0.220 | **10.25** | **0.2794** | **0.1019** | **0.95** |
| MMLU-Professional Psychology (344 items) | $ATLAS_{SE\leq0.1}$ | **0.051** | 58.00 | 0.3224 | 0.1685 | 3.86 |
| | $ATLAS_{SE\leq0.2}$ | 0.166 | 11.53 | 0.1850 | 0.0670 | 1.06 |
| | $ATLAS_{SE\leq0.3}$ | 0.205 | **10.08** | **0.1715** | **0.0645** | **0.95** |
| MMLU-Public Relations (93 items) | $ATLAS_{SE\leq0.1}$ | **0.027** | 93.05 | 1.0000 | 1.0000 | 0.74 |
| | $ATLAS_{SE\leq0.2}$ | 0.048 | 72.97 | 0.7926 | 0.7854 | 0.61 |
| | $ATLAS_{SE\leq0.3}$ | 0.095 | **24.50** | **0.3729** | **0.2633** | **0.31** |
| MMLU-Security Studies (201 items) | $ATLAS_{SE\leq0.1}$ | **0.038** | 100.00 | 0.7247 | 0.4975 | 6.11 |
| | $ATLAS_{SE\leq0.2}$ | 0.119 | 34.74 | 0.3103 | 0.2038 | 2.59 |
| | $ATLAS_{SE\leq0.3}$ | 0.171 | **17.12** | **0.1992** | **0.1103** | **1.30** |
| MMLU-Sociology (166 items) | $ATLAS_{SE\leq0.1}$ | **0.032** | 100.00 | 0.8412 | 0.6024 | 6.02 |
| | $ATLAS_{SE\leq0.2}$ | 0.070 | 72.92 | 0.6391 | 0.5095 | 5.00 |
| | $ATLAS_{SE\leq0.3}$ | 0.137 | **37.22** | **0.3946** | **0.2712** | **2.70** |
| MMLU-US Foreign Policy (90 items) | $ATLAS_{SE\leq0.1}$ | **0.024** | 90.00 | 1.0000 | 1.0000 | 0.72 |
| | $ATLAS_{SE\leq0.2}$ | 0.026 | 89.55 | 0.9950 | 0.9950 | 0.71 |
| | $ATLAS_{SE\leq0.3}$ | 0.063 | **30.48** | **0.4204** | **0.3389** | **0.37** |
| MMLU-Virology (140 items) | $ATLAS_{SE\leq0.1}$ | **0.032** | 100.00 | 0.8644 | 0.7143 | 6.01 |
| | $ATLAS_{SE\leq0.2}$ | 0.081 | 63.64 | 0.5787 | 0.4711 | 4.40 |
| | $ATLAS_{SE\leq0.3}$ | 0.134 | **24.11** | **0.2965** | **0.1926** | **1.84** |
| MMLU-World Religions (137 items) | $ATLAS_{SE\leq0.1}$ | **0.033** | 100.00 | 0.8796 | 0.7299 | 6.05 |
| | $ATLAS_{SE\leq0.2}$ | 0.068 | 87.19 | 0.7734 | 0.6657 | 5.23 |
| | $ATLAS_{SE\leq0.3}$ | 0.134 | **37.56** | **0.4063** | **0.2984** | **2.68** |

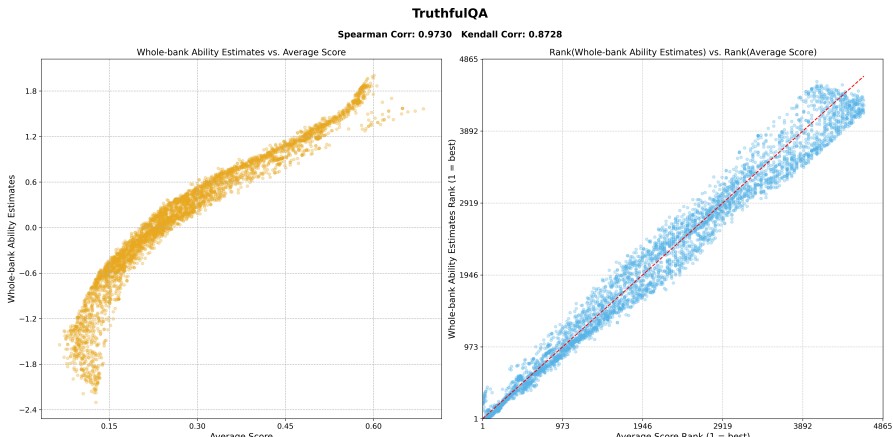

Figure 4: **Comparison of raw average scores and whole-bank ability estimates on TruthfulQA.**
(Left) While average scores compress performance at the extremes, whole-bank ability estimates
reveal clearer separation among both low- and high-performing models, reflecting sensitivity to
item difficulty and discrimination. (Right) Rank comparison shows strong consistency between
the two measures (Spearman $\rho = 0.97$, Kendall $\tau = 0.87$), but ability-based ranking provides
finer resolution, especially in distinguishing weaker and stronger models beyond what raw accuracy
captures.

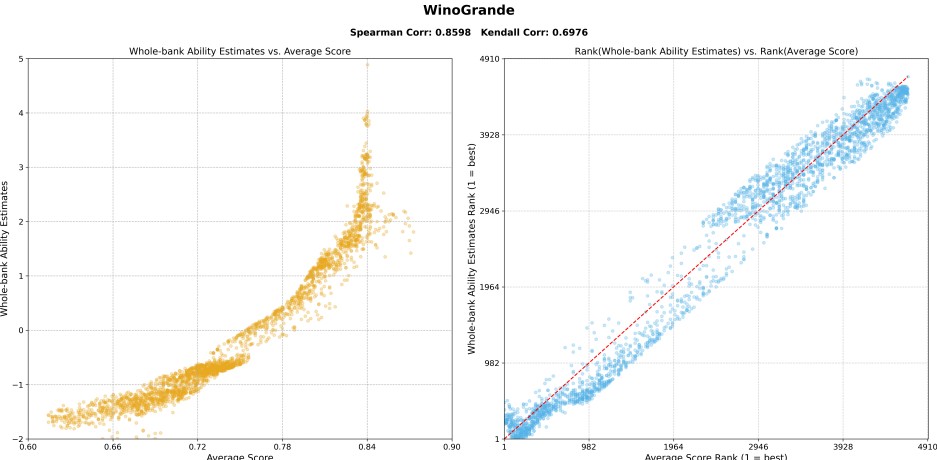

Figure 5: **Comparison of raw average scores and whole-bank ability estimates on WinoGrande.**
(Left) Whole-bank estimates show a non-linear relationship with average score and reveal clearer
separation on high-performing models, highlighting that ability captures relative item difficulty and
provides finer differentiation beyond raw accuracy. (Right) Rank comparison indicates strong but
imperfect alignment (Spearman $\rho = 0.86$, Kendall $\tau = 0.70$), with deviations from the diagonal
reflecting cases where ability-based ranking distinguishes models more effectively than accuracy
alone.

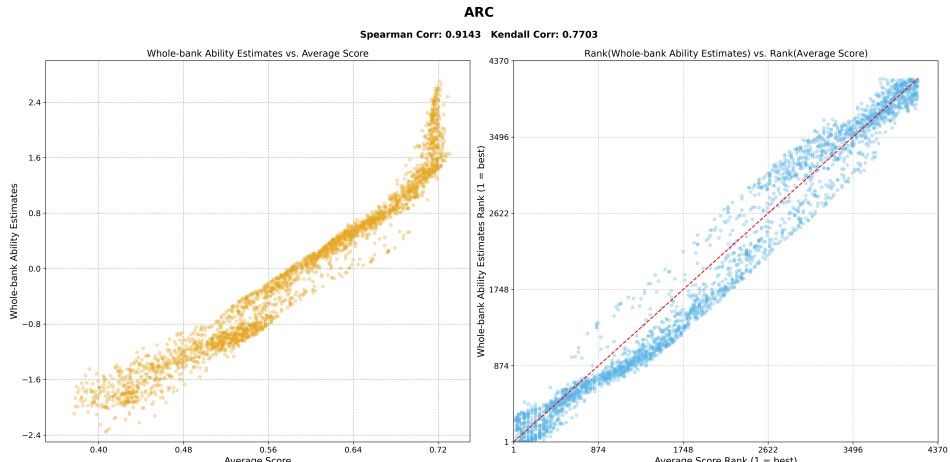

Figure 6: **Comparison of raw average scores and whole-bank ability estimates on ARC.** (Left) Whole-bank estimates exhibit a non-linear relationship with average scores, providing clearer separation on high-performing models by accounting for item difficulty and discrimination. (Right) Rank comparison shows strong but not perfect alignment between the two metrics (Spearman $\rho = 0.91$, Kendall $\tau = 0.77$), with deviations from the diagonal highlighting cases where ability-based ranking offers more informative distinctions than raw accuracy alone.

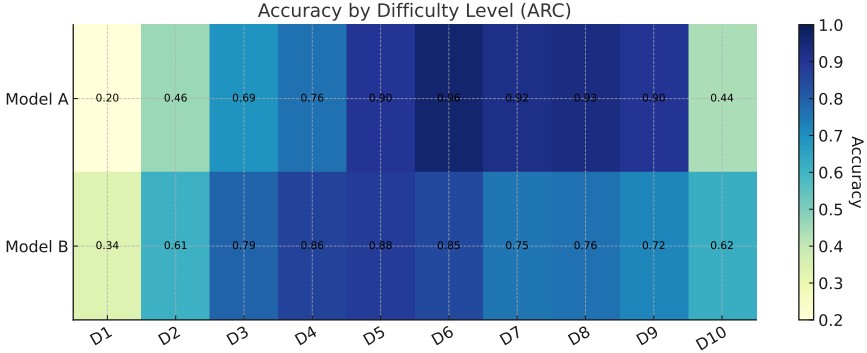

Figure 7: Two models with the similar average accuracy (0.713) and (0.714) on ARC nevertheless receive very different whole-bank ability estimates. Model A (`mera-mix-4x7B`) attains a whole-bank ability rank of 270 because its correct responses are concentrated on more difficult items. In contrast, Model B (`LLaMAAntino-3-ANITA-8B-Inst-DPO-ITA`) is assigned a much lower whole-bank ability rank of 2612, as its successes occur primarily on easier items. This divergence shows how IRT-based ability estimation can distinguish models that appear identical under raw accuracy by accounting for item difficulty.

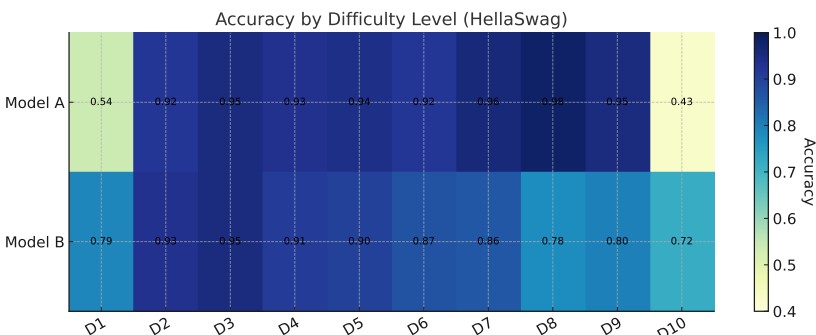

Figure 8: Two models with the same average accuracy (0.853) on HellaSwag nevertheless receive very different whole-bank ability estimates. Model A (`supermario_v1`) attains a whole-bank ability rank of 347 because its correct responses are concentrated on more difficult items. In contrast, Model B (`contaminated_proof_7b_v1.0_safetensor`) is assigned a much lower whole-bank ability rank of 3074, as its successes occur primarily on easier items. This divergence shows how IRT-based ability estimation can distinguish models that appear identical under raw accuracy by accounting for item difficulty.

