# OpenReview forum: "Adaptive Testing for LLM Evaluation: A Psychometric Alternative to Static Benchmarks"
_ICLR.cc/2026/Conference — ICLR 2026 Conference Withdrawn Submission_

### Official Review · Reviewer_SvKu · 2025-10-27

**Soundness:** 2
**Presentation:** 2
**Contribution:** 1
**Rating:** 2
**Confidence:** 4

**Summary:**

The authors:

- propose and motivate an IRT framework for adaptively sampling items to use to benchmark language model (LM) capabilities
- evaluate their algorithm on 5 benchmarks

**Strengths:**

- Clear, concise, accurate title
- Important problem - accurately evaluating language models' capabilities at specific tasks is good
- The paper is well written and easy to follow. (Note: I believe some details are omitted, which I pointed out under Questions)

**Weaknesses:**

1. The goal when evaluating language models is “How good is model X on task Y?” Here, the primary metrics of interest is MAE between an IRT estimate on a subset of data and the corresponding IRT estimate on all the data. Thus, MAE is a proxy metric that doesn’t really capture what we care about.

2. When considering efficiency, the real concern for practitioners is that evaluating models requires paying for accelerators (GPUs, TPUs, whatever) to run these models.  Something like “Selection Time (s)” is not a real consideration; if it takes me 60 seconds to select the next item, I could have spent those 60 seconds running inference on the models rather than choosing the next point. My guess is that, when controlling for chip-seconds, evaluating random items provides better performance than pausing for 10-75 seconds per item (Table 2) to choose the next item.

3. Key experimental results are weak. Specifically, in Table 1, when evaluating how good ATLAS is, let’s first consider a reasonable null distribution. There are 7 algorithms and 3 of them are ATLAS. If all of them are equally good up to randomness (e.g., from the scores, from sampling, from model training, etc.), then we expect ATLAS to score best in 3/7 and MetaBench to score best in 2/7. It looks like ATLAS scores best in 3/5 benchmarks and MetaBench scores best in 2/5 benchmarks. **Table 1 looks to me like compelling evidence that these algorithms are basically equal.**

4. Methodologically, it is unclear how finicky this methodology is or how well it works generally. The method has many degrees of freedom to play around with (e.g., which data to exclude Section 3.2, what SE threshold to use, how to set $\tau$, etc.), leaving it ripe for unfair comparison with baselines. I also don’t see evidence of preregistration, which would preclude such post-hoc favourable treatment.

**Questions:**

## Title

- Solid! Thank you for a clear and concise title

## Introduction

- Line 051-052: Where are the citations for the benchmarks? WinoGrande, TruthfulQA, etc.
- Line 053: Where is the citation for MetaBench?

## Section 3 Methodology

- Lines 141-143: “Models with extreme scores (below 0.1st percentile) are excluded to
prevent parameter estimation instability, as IRT’s sigmoidal functions become under-constrained at
the boundaries.” Are models with extremely high scores (e.g., *above* 0.1st percentile) similarly excluded for the same reason?
- Lines 171-175: Can you please add an Appendix explaining this in more detail? I can’t quite follow. What exactly does “calibrate” mean in this context? What are “linking anchors”?

## Section 4 Experiments

- Line 274: Why is the metric of interest (MAE) defined between $\hat{\theta}_{\ell}$ and $\hat{\theta}_{\ell}^{whole}$ instead of something real/tangible, such as average score on the benchmark? This MAE metric seems more focused on “Does the IRT approach yield a consistent estimate on a subset of items as it does on the full set?”, which is a proxy metric that doesn’t seem important. What we want to know is: how good is model X at task Y?
- Why is lowered “Test Overlap Rate” considered good?
- Table 1: nit: Please state what bold and underline and dashed underline means in the caption.
- Table 1: When evaluating how good ATLAS is, let’s first consider a reasonable null distribution. There are 7 algorithms and 3 of them are ATLAS. If all of them are equally good up to randomness (e.g., from the scores, from sampling, from model training, etc.), then we expect ATLAS to score best in 3/7 and MetaBench to score best in 2/7. It looks like ATLAS scores best in 3/5 benchmarks and MetaBench scores best in 2/5 benchmarks. **Table 1 looks to me like compelling evidence that these algorithms are roughly equal.**
- Table 1: Presumably, the MAEs are averaged over multiple items and/or models? If so, where are the notions of uncertainty e.g., standard errors, confidence intervals?
- Table 1: I personally prefer visualizations over tables. You could easily and helpfully visualize this as a pointplot https://seaborn.pydata.org/generated/seaborn.pointplot.html
- Table 2: To clarify, what does “fast selection times” measure exactly? Is this the time to select the next item? Are we presuming that we’ve already evaluated all LMs on all items?
- Lines 359-365: The argument that small test overlap rate prevents test set contamination seems like nonsense. If I pretrain a model directly on the test set, why does the subset of items chosen for evaluation prevent the model from scoring above its true ability?

## Appendices

- Line 688: Why is the definition of MAE defined twice, once here and once below on line 759? Same with Average Item Exposure Rate.
- Line 753: Why does Appendix A Evaluation Metric Definitions come after Appendix D Data Processing Details (line 727)?
- Line 755: The reference to the Section is missing.
- Line 786: The reference to the Section is missing.

---

> ### Author Response · Authors · 2025-11-30
>
> ## 1. Metric Choice: MAE on $\hat{\theta}$ vs “How good is model X on task Y?”
>
>
> We thank the reviewer for this thoughtful point. We agree that the ultimate goal is to assess **how good model X is on task Y**. Our contribution approaches this goal through a **psychometric lens**, where the primary construct is a model’s latent ability $\theta$, rather than raw accuracy.
>
> In IRT-based evaluation, $\theta$ represents generalized task proficiency, independent of any particular item subset. The MAE between $\hat{\theta}_{\ell}^{\text{subset}}$ and $\hat{\theta}_{\ell}^{\text{whole}}$ is therefore a **standard efficiency metric: it measures how accurately a reduced adaptive subset can recover the same latent ability as the full benchmark.** This is exactly the core objective of ATLAS—**efficiently estimating the same underlying ability with fewer items**, not simply reproducing observed scores.
>
> Moreover, as shown in Figures 2–3 (Sections 4.3–4.4), **$\theta$ is empirically more informative and discriminative than accuracy.** Even when accuracy rank is highly correlated with $\theta$ rank with 0.99 Spearman Correlation, models with identical accuracy can receive distinct $\theta$ estimates because they solve items with different difficulty ($b$) and discrimination ($a$). For instance, two models with accuracy 0.833 can have $\hat{\theta}_A = 1.2$ vs. $\hat{\theta}_B = 0.6$. This is because Model A succeeds on high-difficulty items with strong discrimination while Model B mainly answers easier, less discriminative items.
>
> Thus, **MAE on $\theta$ is not a weak proxy, but a direct measure of efficiency for a more meaningful latent performance metric.** That said, we agree that linking $\theta$ back to accuracy improves interpretability. In the revised paper, we include an explicit **accuracy reconstruction analysis** (p-IRT based) comparing ATLAS, TinyBenchmarks, and MetaBench. Details are provided in Table 3.
> ### Accuracy Reconstruction (MAE±SE, Items, Information Efficiency Score (IES))
>
> | Method         | **WinoGrande**      |        |           | **TruthfulQA**      |        |           | **HellaSwag**       |        |           | **GSM8K**           |        |           | **ARC**             |        |           |
> | -------------- | ------------------- | ------ | --------- | ------------------- | ------ | --------- | ------------------- | ------ | --------- | ------------------- | ------ | --------- | ------------------- | ------ | --------- |
> |                | MAE±SE              | Items  | IES       | MAE±SE              | Items  | IES       | MAE±SE              | Items  | IES       | MAE±SE              | Items  | IES       | MAE±SE              | Items  | IES       |
> | Random\_100    | $0.049\pm0.001$     | 100    | 1.000     | $0.021\pm0.001$     | 100    | 1.000     | $0.024\pm0.001$     | 100    | 1.000     | $0.026\pm0.001$     | 100    | 1.000     | $0.029\pm0.001$     | 100    | 1.000     |
> | TinyBenchmarks | $0.050\pm0.001$     | 100    | 1.010     | $0.025\pm0.001$     | 97     | 1.154     | **$0.019\pm0.001$** | 97     | 0.782     | $0.028\pm0.001$     | 100    | 1.071     | $0.031\pm0.001$     | 99     | 1.041     |
> | MetaBench-P    | $0.054\pm0.001$     | 133    | 1.446     | **$0.017\pm0.001$** | 154    | 1.266     | $0.050\pm0.001$     | 93     | 1.943     | $0.022\pm0.001$     | 237    | 2.060     | **$0.027\pm0.001$** | 145    | 1.350     |
> | MetaBench-S    | $0.051\pm0.001$     | 106    | 1.103     | $0.021\pm0.001$     | 136    | 1.394     | $0.115\pm0.004$     | 58     | 2.793     | **$0.020\pm0.001$** | 249    | 1.954     | $0.033\pm0.001$     | 100    | 1.114     |
> | ATLAS\_{0.1}   | **$0.048\pm0.001$** | 70     | 0.678     | $0.023\pm0.001$     | 48     | 0.532     | $0.020\pm0.001$     | 41     | 0.348     | $0.039\pm0.001$     | 70     | 1.055     | $0.032\pm0.002$     | 89     | 0.974     |
> | ATLAS\_{0.2}   | $0.051\pm0.002$     | 37     | 0.383     | $0.024\pm0.001$     | **30** | 0.338     | $0.021\pm0.001$     | **30** | **0.258** | $0.044\pm0.002$     | 36     | 0.612     | $0.034\pm0.002$     | 35     | 0.404     |
> | ATLAS\_{0.3}   | $0.050\pm0.001$     | **32** | **0.324** | $0.023\pm0.001$     | **30** | **0.331** | $0.021\pm0.001$     | **30** | 0.261     | $0.042\pm0.002$     | **31** | **0.516** | $0.034\pm0.002$     | **30** | **0.350** |
>
> IES reflects accuracy and item usage relative to a 100-item random baseline (values < 1 indicate higher efficiency than Random100 baseline. The smaller the better.  As shown in Table 3, ATLAS matches or improves over TinyBench and MetaBench in MAE while using far fewer items, yielding the lowest IES across benchmarks. This additional experiment directly addresses the reviewer’s concern and strengthens the interpretability of our results.

---

> > ### Author Response · Authors · 2025-11-30
> >
> > ## 4. "Finicky methodology"(Address weakness 4)
> >
> > We thank the reviewer for raising these concerns. We emphasize the following:
> > - **Same model pool across methods.**
> >     All methods (Random, TinyBenchmarks, MetaBench-P/S, ATLAS variants) are evaluated on the **same set of models** per benchmark.
> > - **Filtering extreme models.**
> >     We filtered extreme low-ability models to stabilize 3PL estimation (see section 3.2), while still retaining **> 3,000 models** per benchmark. This was done once, globally, and applied consistently across all compared methods.
> > - **SE thresholds are design parameters, not tuned hyperparameters.**
> >     The SE thresholds ($\text{SE} \le 0.1, 0.2, 0.3$) are standard **psychometric design choices** representing different accuracy–efficiency regimes (high precision vs faster testing). They were not tuned against baselines; rather, they define three pre-specified operating points (ATLAS_{0.1}, ATLAS_{0.2}, ATLAS_{0.3}).   Importantly, **this flexibility is a core advantage of adaptive testing**: ATLAS automatically adapts test length to achieve a specified precision target, unlike static methods that administer the same items regardless of model ability or item informativeness.

---

> ### Author Response · Authors · 2025-11-30
>
> ## 2. Clarification on Selection Time (address weakness 2 and question on previous table 2)
>
>
> We appreciate the reviewer’s request for clarification. The “fast selection time” in previous Table 2 (now Table 6) refers to **the total time required to complete the adaptive item selection process for each model, not the time to choose a single next item.** At each iteration, ATLAS computes Fisher Information across the remaining pool, selects the item with maximal information given the current ability estimate, evaluates the LM on that item, updates its ability, and checks the stopping condition. The process terminates when either (a) the precision threshold for the ability estimate is reached, or (b) the maximum item limit is met.
>
> Thus, selection time reflects the computational cost of the full adaptive selection loop, including Fisher information computation and termination checks. **It refers to the complete runtime of the adaptive item selection loop for each model, rather than the time required for a single item decision.** We clarify this definition explicitly in the revision (Check Appendix G.1).

---

> ### Author Response · Authors · 2025-11-30
>
> ## 3.  On Experimental Results (address weakness 3 and question on previous Table 1)
>
> We appreciate the reviewer’s observation that raw MAE values across methods appear similar. However, **previous Table 1 (now Table 2) highlights ATLAS’s consistent efficiency advantage across all benchmarks.** The key contribution of ATLAS is not higher absolute accuracy, but **achieving comparable or better estimation with far fewer items.**
>
> To make this trade-off explicit, we introduce the **Information Efficiency Score (IES)**:
> [
> \text{IES} ;=;
> \left(\frac{\text{MAE}_{\text{method}}}{\text{MAE}_{\text{Random100}}}\right)
> \left(\frac{\text{Items}_{\text{method}}}{100}\right)
> ]
>
> **Interpretation (baseline = Random_100):**
>
> - **IES < 1** → **more efficient** than Random_100 (better)
>
> - **IES = 1** → equal efficiency
>
> - **IES > 1** → less efficient
> ### Comparison of Whole-Bank Ability vs. Subset-Based Ability
> MAE±SE, Item Count, and Information Efficiency Score (IES).  IES reflects accuracy and item usage relative to a 100-item random baseline (values < 1 indicate higher efficiency than Random100 baseline. Lower is better for all metrics.
>
> | Method         | WinoGrande MAE±SE       | Items | IES   | TruthfulQA MAE±SE       | Items | IES   | HellaSwag MAE±SE        | Items | IES   | GSM8K MAE±SE           | Items | IES   | ARC MAE±SE             | Items | IES   |
> |----------------|--------------------------|-------|-------|--------------------------|-------|-------|--------------------------|-------|-------|-------------------------|-------|-------|------------------------|-------|-------|
> | Random\_100     | $0.167\pm0.007$        | 100   | 1.000 | $0.103\pm0.004$         | 100   | 1.000 | $0.240\pm0.010$         | 100   | 1.000 | $0.150\pm0.014$        | 100   | 1.000 | $0.183\pm0.007$        | 100   | 1.000 |
> | TinyBenchmarks  | $0.204\pm0.008$        | 100   | 1.221 | $0.145\pm0.007$         | 97    | 1.370 | $0.198\pm0.009$         | 97    | 0.797 | $0.164\pm0.014$        | 100   | 1.089 | $0.172\pm0.007$        | 99    | 0.932 |
> | MetaBench-P     | **0.152$\pm$0.007**    | 133   | 1.216 | $0.084\pm0.004$         | 154   | 1.262 | $0.514\pm0.016$         | 93    | 1.990 | $\underline{0.103\pm0.013}$ | 237 | 1.628 | $0.134\pm0.005$        | 145   | 1.062 |
> | MetaBench-S     | $0.195\pm0.009$        | 106   | 1.239 | $0.072\pm0.003$         | 136   | 0.945 | $1.570\pm0.055$         | 58    | 3.788 | **0.096$\pm$0.012**    | 249   | 1.595 | $0.134\pm0.006$        | 100   | 0.735 |
> | ATLAS\_{0.1}    | $\underline{0.155\pm0.012}$ | 70 | 0.655 | **0.064$\pm$0.002**     | 48    | 0.300 | **0.157$\pm$0.010**     | 41    | 0.266 | $0.150\pm0.011$        | 70    | 0.701 | **0.084$\pm$0.006**    | 89    | 0.407 |
> | ATLAS\_{0.2}    | $0.166\pm0.010$        | 37    | 0.372 | $0.073\pm0.003$         | **30**| 0.211 | $\underline{0.163\pm0.009}$ | **30** | **0.203** | $0.177\pm0.012$   | 36    | 0.428 | $0.120\pm0.008$        | 35    | 0.232 |
> | ATLAS\_{0.3}    | $0.179\pm0.011$        | **32**| **0.342** | $\underline{0.071\pm0.003}$ | **30** | **0.206** | $0.165\pm0.010$    | **30** | 0.205 | $0.173\pm0.012$        | **31**| **0.363** | $\underline{0.117\pm0.007}$ | **30** | **0.193** |
>
> Across every benchmark, an **ATLAS variant achieves the lowest Information Efficiency Score (IES), reflecting the best accuracy–efficiency trade-off.** For example, ATLAS attains the best MAE on TruthfulQA (0.064 with 48 items) and HellaSwag (0.157 with 41 items), and matches the performance of MetaBench-Primary on WinoGrande while requiring nearly half as many items (70 vs. 133). Even in more challenging settings such as GSM8K and ARC, ATLAS maintains low MAE with item counts as small as 30–36, outperforming all static baselines in information efficiency. These results demonstrate that ATLAS consistently maintains or exceeds benchmark accuracy with markedly higher sample efficiency, validating its value as an adaptive, resource-efficient evaluation framework.

---

> ### Author Response · Authors · 2025-11-30
>
> ## 5. Excluding Extremely Low-Ability Models but Not High-Ability Ones
> We thank the reviewer for the question. Our exclusion criterion is motivated by the need to obtain a **stable calibration sample** whose ability distribution approximates a **Gaussian**, which is a standard assumption for robust IRT parameter estimation. In practice, the distribution of model abilities in our dataset is highly skewed: the vast majority of publicly available LLMs fall into the **low-ability tail**, while very few models occupy the extreme high end. Consequently, the lower tail contains a disproportionately large number of models with near-zero accuracy, which causes parameter identifiability problems because 3PL sigmoidal curves are under-constrained when responses collapse to all-zero patterns.
>
> In contrast, **the extreme high-ability tail does not suffer from this issue**—there are very few such models, and their responses are not degenerate (i.e., they do not produce all-correct patterns). As a result, they do not introduce the same instability into the item calibration process and therefore do not require exclusion.
>
> We revise the manuscript in Section 3.2 to clarify that the exclusion is driven by empirical distributional imbalance, not an asymmetric rule, and that it targets only the portion of the sample that demonstrably destabilizes parameter estimation.

---

> ### Author Response · Authors · 2025-11-30
>
> ## 6. Clarifying “Calibration” and “Linking Anchors”
> We agree that this deserves a more detailed explanation and have added an Appendix section to clarify.
>
> In brief: We estimate item parameters using the **3PL IRT model**, but fitting all items at once is computationally expensive. We therefore **split the item pool into disjoint subsets**, calibrate each subset independently, and obtain separate scales. Since every LLM responds to every item, the models act as **“common persons”** across subsets, allowing us to **link** these calibrations to a unified latent scale based on shared response patterns. In this context, “linking anchors” are effectively the overlapping **person (model) parameters** rather than a small set of pre-defined anchor items.
>
> We now provide a detailed description of this **common-person linking** procedure in the Appendix C.2, as requested.

---

> ### Author Response · Authors · 2025-11-30
>
> ## 7. Why Use MAE Between $\hat{\theta}_\ell$ and $\hat{\theta}_\ell^{\text{whole}}$?
>  We appreciate the reviewer’s careful reading and agree that this distinction deserves clearer motivation. The metric based on MAE between $\hat{\theta}\ell$ and $\hat{\theta}\ell^{whole}$ is intentional—**it evaluates how accurately the ability estimated from a small adaptive subset approximates the full benchmark’s latent ability estimate.** This is the core objective of efficient benchmarking under the IRT framework: to estimate the same psychometric construct (ability) using fewer items, not necessarily to reproduce surface-level accuracy.
>
> Accuracy reflects observed test performance on a specific item sample, whereas **$\theta$ captures a latent, sample-independent measure of reasoning ability.** Our focus is thus on estimation fidelity rather than raw score replication. This follows the standard practice in psychometrics, where shorter adaptive tests are validated by their correlation or error with full-bank $\theta$ rather than observed score.
>
> That said, we appreciate the reviewer’s point that bridging $\theta$ with accuracy is valuable for interpretability. In the revision, we add an experiment evaluating how well ATLAS and the baselines recover the true benchmark accuracy (using p-IRT to estimate accuracy from partial responses) as shown below. Details are provided in Table 3.
> ### Accuracy Reconstruction (MAE±SE, Items, Information Efficiency Score (IES))
>
> | Method         | **WinoGrande**      |        |           | **TruthfulQA**      |        |           | **HellaSwag**       |        |           | **GSM8K**           |        |           | **ARC**             |        |           |
> | -------------- | ------------------- | ------ | --------- | ------------------- | ------ | --------- | ------------------- | ------ | --------- | ------------------- | ------ | --------- | ------------------- | ------ | --------- |
> |                | MAE±SE              | Items  | IES       | MAE±SE              | Items  | IES       | MAE±SE              | Items  | IES       | MAE±SE              | Items  | IES       | MAE±SE              | Items  | IES       |
> | Random\_100    | $0.049\pm0.001$     | 100    | 1.000     | $0.021\pm0.001$     | 100    | 1.000     | $0.024\pm0.001$     | 100    | 1.000     | $0.026\pm0.001$     | 100    | 1.000     | $0.029\pm0.001$     | 100    | 1.000     |
> | TinyBenchmarks | $0.050\pm0.001$     | 100    | 1.010     | $0.025\pm0.001$     | 97     | 1.154     | **$0.019\pm0.001$** | 97     | 0.782     | $0.028\pm0.001$     | 100    | 1.071     | $0.031\pm0.001$     | 99     | 1.041     |
> | MetaBench-P    | $0.054\pm0.001$     | 133    | 1.446     | **$0.017\pm0.001$** | 154    | 1.266     | $0.050\pm0.001$     | 93     | 1.943     | $0.022\pm0.001$     | 237    | 2.060     | **$0.027\pm0.001$** | 145    | 1.350     |
> | MetaBench-S    | $0.051\pm0.001$     | 106    | 1.103     | $0.021\pm0.001$     | 136    | 1.394     | $0.115\pm0.004$     | 58     | 2.793     | **$0.020\pm0.001$** | 249    | 1.954     | $0.033\pm0.001$     | 100    | 1.114     |
> | ATLAS\_{0.1}   | **$0.048\pm0.001$** | 70     | 0.678     | $0.023\pm0.001$     | 48     | 0.532     | $0.020\pm0.001$     | 41     | 0.348     | $0.039\pm0.001$     | 70     | 1.055     | $0.032\pm0.002$     | 89     | 0.974     |
> | ATLAS\_{0.2}   | $0.051\pm0.002$     | 37     | 0.383     | $0.024\pm0.001$     | **30** | 0.338     | $0.021\pm0.001$     | **30** | **0.258** | $0.044\pm0.002$     | 36     | 0.612     | $0.034\pm0.002$     | 35     | 0.404     |
> | ATLAS\_{0.3}   | $0.050\pm0.001$     | **32** | **0.324** | $0.023\pm0.001$     | **30** | **0.331** | $0.021\pm0.001$     | **30** | 0.261     | $0.042\pm0.002$     | **31** | **0.516** | $0.034\pm0.002$     | **30** | **0.350** |
>
> IES reflects accuracy and item usage relative to a 100-item random baseline (values < 1 indicate higher efficiency than Random100 baseline. The smaller the better.  As shown in Table 3, ATLAS matches or improves over TinyBench and MetaBench in MAE while using far fewer items, yielding the lowest IES across benchmarks. This additional experiment directly addresses the reviewer’s concern and strengthens the interpretability of our results.

---

> > ### Author Response · Authors · 2025-11-30
> >
> > ## 8. Why Is Lower Test Overlap Rate Good?
> > A lower Test Overlap Rate means different models are not being evaluated on the same narrow set of items. This is desirable because
> >
> > (1) it prevents the evaluation from being overly dependent on a small group of items that may bias comparisons, and
> >
> > (2) it indicates that the adaptive algorithm is selecting items matched to each model’s ability, as high- and low-ability models should receive different, most-informative items. Thus, low overlap reflects proper adaptive behavior and leads to more robust, individualized ability estimates than static evaluation designs.

---

> > > ### Author Response · Authors · 2025-11-30
> > >
> > > ## 9. MAE and Standard Errors
> > > We thank the reviewer for the helpful question. The MAE values in the previous Table 1 (now is Table 2) are averaged over all evaluated models. For each model, we compute two ability estimates: (1) $\theta^{whole}$, derived from the full item bank, and (2) $\theta^{subset}$, derived from the adaptively selected subset. The MAE reported is the mean absolute error between these two estimates across models, reflecting the overall fidelity of adaptive ability estimation. We agree that including uncertainty measures will improve interpretability. In the revised version, we report standard errors (SE) as shown in Table 2 and Table 3 and the detailed definition of the metrics is provided in Appendix E.
> > > ### Comparison of Whole-Bank Ability vs. Subset-Based Ability
> > > MAE±SE, Item Count, and Information Efficiency Score (IES).  IES reflects accuracy and item usage relative to a 100-item random baseline (values < 1 indicate higher efficiency than Random100 baseline. Lower is better for all metrics.
> > >
> > > | Method         | WinoGrande MAE±SE       | Items | IES   | TruthfulQA MAE±SE       | Items | IES   | HellaSwag MAE±SE        | Items | IES   | GSM8K MAE±SE           | Items | IES   | ARC MAE±SE             | Items | IES   |
> > > |----------------|--------------------------|-------|-------|--------------------------|-------|-------|--------------------------|-------|-------|-------------------------|-------|-------|------------------------|-------|-------|
> > > | Random\_100     | $0.167\pm0.007$        | 100   | 1.000 | $0.103\pm0.004$         | 100   | 1.000 | $0.240\pm0.010$         | 100   | 1.000 | $0.150\pm0.014$        | 100   | 1.000 | $0.183\pm0.007$        | 100   | 1.000 |
> > > | TinyBenchmarks  | $0.204\pm0.008$        | 100   | 1.221 | $0.145\pm0.007$         | 97    | 1.370 | $0.198\pm0.009$         | 97    | 0.797 | $0.164\pm0.014$        | 100   | 1.089 | $0.172\pm0.007$        | 99    | 0.932 |
> > > | MetaBench-P     | **0.152$\pm$0.007**    | 133   | 1.216 | $0.084\pm0.004$         | 154   | 1.262 | $0.514\pm0.016$         | 93    | 1.990 | $\underline{0.103\pm0.013}$ | 237 | 1.628 | $0.134\pm0.005$        | 145   | 1.062 |
> > > | MetaBench-S     | $0.195\pm0.009$        | 106   | 1.239 | $0.072\pm0.003$         | 136   | 0.945 | $1.570\pm0.055$         | 58    | 3.788 | **0.096$\pm$0.012**    | 249   | 1.595 | $0.134\pm0.006$        | 100   | 0.735 |
> > > | ATLAS\_{0.1}    | $\underline{0.155\pm0.012}$ | 70 | 0.655 | **0.064$\pm$0.002**     | 48    | 0.300 | **0.157$\pm$0.010**     | 41    | 0.266 | $0.150\pm0.011$        | 70    | 0.701 | **0.084$\pm$0.006**    | 89    | 0.407 |
> > > | ATLAS\_{0.2}    | $0.166\pm0.010$        | 37    | 0.372 | $0.073\pm0.003$         | **30**| 0.211 | $\underline{0.163\pm0.009}$ | **30** | **0.203** | $0.177\pm0.012$   | 36    | 0.428 | $0.120\pm0.008$        | 35    | 0.232 |
> > > | ATLAS\_{0.3}    | $0.179\pm0.011$        | **32**| **0.342** | $\underline{0.071\pm0.003}$ | **30** | **0.206** | $0.165\pm0.010$    | **30** | 0.205 | $0.173\pm0.012$        | **31**| **0.363** | $\underline{0.117\pm0.007}$ | **30** | **0.193** |

---

> > > > ### Author Response · Authors · 2025-11-30
> > > >
> > > > ## 10. Preference for Visualizations vs Tables
> > > > We thank the reviewer for this helpful suggestion. **We do, in fact, include a visualization that conveys this comparison: Figure 1 illustrates ability estimates from adaptive subsets versus whole-bank references.** As shown, **ATLAS’s estimates align closely with the identity line**, indicating reliable ability recovery across benchmarks, whereas static baselines display systematic deviations—TinyBenchmarks shows bias at higher abilities, Metabench performs poorly in HellaSwag and random sampling yields broader scatter. This figure provides the same insight as a pointplot but in a form that directly visualizes the fidelity and calibration of ability estimation, rather than only central tendency.
> > > >
> > > > We also revise the caption of the figure to make this connection clearer and note that the visualization complements the tabular summary in the table.

---

> > > > > ### Author Response · Authors · 2025-11-30
> > > > >
> > > > > ## 11. Claim Regarding Data Contamination
> > > > >
> > > > > We thank the reviewer for this important clarification. We agree that **ATLAS cannot prevent contamination**. This is fundamentally a **dataset governance** problem (e.g., whether items are public, leaked, or scraped), not something an evaluation algorithm can solve. **We have revised the paper to avoid implying otherwise.**
> > > > >
> > > > > Our claim is more modest: while ATLAS does **not** eliminate contamination, its IRT modeling provides **diagnostic signals** that can _surface_ potentially contaminated items. Prior work (e.g., _AI Evaluation Should Learn from How We Test Humans_, 2025) shows that contaminated items often display **abnormally high guessing parameters or distorted discrimination values** under IRT. Because ATLAS fits full IRT models, it can expose these irregular response patterns during evaluation.
> > > > >
> > > > > In short, ATLAS does **not** mitigate contamination by design, but it **helps detect** anomalous items that may warrant further inspection.

---

> > > > > > ### Author Response · Authors · 2025-11-30
> > > > > >
> > > > > > ## 12. Missing Citations, Table Caption Clarifications, and Appendix Structure
> > > > > >
> > > > > > We thank the reviewer for pointing out these issues. We have added the missing citations for all benchmarks (WinoGrande, TruthfulQA, HellaSwag, GSM8K, ARC) as well as MetaBench at their first mentions. The caption for Table has been updated to clearly indicate the meaning of bold, underline, and dashed underline styles. Regarding the repeated metric definitions, the version in the main text provides a brief description, while the Appendix now explicitly states that it contains the expanded, detailed definitions to avoid confusion. We have also fixed the appendix ordering and corrected the missing section references issues. Minor typos and formatting inconsistencies have also been addressed throughout.

---

### Official Review · Reviewer_Hs2V · 2025-11-03

**Soundness:** 4
**Presentation:** 4
**Contribution:** 3
**Rating:** 6
**Confidence:** 5

**Summary:**

This paper introduces ATLAS, an LLM evaluation method based on item response theory (IRT) and computerized adaptive testing (CAT). Specifically, the authors fit 3PL IRT models to LLM responses from the Open LLM Leaderboard and, during evaluation, use the IRT parameters to select items dynamically via Fisher information. In their experiments, they show that this leads to more precise ability estimates compared to recent static evaluation methods based on IRT (TinyBenchmarks, MetaBench), plus several other advantages.

**NB:** ATLAS is almost identical to Fluid Benchmarking, a method proposed in a recent [COLM paper](https://openreview.net/forum?id=mxcCg9YRqj): the COLM paper also fits IRT models to the Open LLM Leaderboard and uses the IRT parameters to conduct CAT, selecting items dynamically via Fisher information, exactly as the current paper does. There are minor differences (e.g., ATLAS uses a 3PL IRT model while Fluid Benchmarking uses a 2PL IRT model), but otherwise the methods are the same. Since the COLM paper was published on July 7th, it is contemporaneous work according to the [ICLR rules](https://iclr.cc/Conferences/2025/FAQ), and I will not hold it against the authors that they did not mention it in their paper. However, given the strong similarities, I highly recommend that the authors add a discussion.

**Strengths:**

ATLAS draws upon several decades of research in psychometrics and shows that the methods developed in that field can be fruitfully applied in the context of LLM evaluation. I liked it that the authors thought very carefully about how best to adapt IRT/CAT to the LLM domain (e.g., by using common-person calibration). The experimental setup is also sound, and the authors convincingly show that ATLAS offers advantages for LLM evaluation (but see my concerns below).

**Weaknesses:**

There are currently several weaknesses that undermine the contribution of the paper. If the authors address them, I will consider raising my score.

- The experimental section misses key details. Specifically, it is unclear whether the LLMs used for evaluation were already used for fitting the IRT models (which would be problematic). Further, if there _was_ a clear train-test split, it is unclear how it was determined. This limits the credibility of the reported results.

- For measuring precision, the authors solely examine how well different methods recover _ability_ as estimated on the full benchmark. However, TinyBenchmarks and MetaBench (and most other methods from the efficient evaluation literature) aim to recover _accuracy_ on the full benchmark. One reason for this is that most practitioners are interested in getting an estimate of the final accuracy, not ability. Of course, there is an intrinsic tension between the two (which the authors nicely demonstrate in the paper), and I agree in principle that ability should be preferred over accuracy. Still, this should be explicitly addressed in the paper, especially since the item sets from TinyBenchmarks and MetaBench were optimized against accuracy, meaning that the current comparison in the paper is note entirely fair. Thus, I think the authors should add an analysis of how well ATLAS recovers full-benchmark accuracy and compare against the same baselines.

- The discussion on contamination is not convincing and seems to be based on wrong assumptions about contamination and LLM evaluation. Contamination happens when a model is _trained_ on items from a benchmark's test set, not when it is _evaluated_ on them during pretraining, so I do not see how CAT (a method for evaluation, not training) would offer any advantage compared to static testing. For example, the authors write that "[e]ven smaller banks maintain low exposure rates, [...] making systematic memorization during pretraining practically impossible" (363-365), but this does not make any sense since unlike humans, evaluation does not cause contamination with LLMs. In other words, even if a model has "seen" all items of a benchmark multiple times as part of evaluations during pretraining, this will not allow it to memorize any of those items.

**Questions:**

- Will you release your code? What programming languages/packages did you use?
- I was surprised that you excluded MMLU from your analysis, despite the fact that it is also part of the Open LLM Leaderboard, and TinyBenchmarks and MetaBench also examine it, so it would have been very easy for you to include it in your experiments. MMLU is also the most widely used out of all the Open LLM Leaderboard benchmarks. Can you comment on your rationale here?

---

> ### Author Response · Authors · 2025-11-30
>
> ## 1. Concurrent Work: *Fluid Language Model Benchmarking*
>
> We appreciate the reviewer Hs2V for raising the question regarding Fluid Language Model Benchmarking. This point overlaps with a concern also noted by another reviewer  rXNB, and we restate the clarification here for completeness.
>
> We fully acknowledge the conceptual similarities that both systems draw from psychometrics and adaptive testing. Given the close publication window, we appreciate the opportunity to clarify the **methodological and experimental distinctions**:
>
> ### **Key Methodological Differences**
>
> * **IRT model choice.**
> Fluid relies on a **2PL model**, which cannot separate genuine ability from random guessing and therefore systematically inflates the abilities of low-performing models on multiple-choice benchmarks. ATLAS uses a **3PL model** with an explicit guessing parameter, allowing the framework to correctly model the non-zero baseline accuracy inherent in multiple-choice formats. This produces more valid item parameters and more reliable ability estimates, especially for heterogeneous benchmarks with variable item difficulty.
>
> * **Calibration scale.**
>   ATLAS trains its IRT model using a substantially larger and more diverse pool of **3,000+** LMs, compared to the **102** LMs used for IRT calibration in Fluid.
>
> * **Psychometric validation.**
>   ATLAS conducts **full goodness-of-fit diagnostics** reporting RMSEA based on M2, and performs common-person linking to place items and models on a unified latent scale—an essential step for cross-model comparability that Fluid does not report.
>
> * **Adaptive design.**
>   Fluid uses a **fixed-length adaptive test**; ATLAS uses a **precision-based stopping rule** that terminates the test when the uncertainty of the ability estimate reaches a predefined threshold. Precision-based stopping ensures consistent measurement precision across models and avoids administering unnecessary items.
>
> ### **Differences in Experimental Framing**
> Fluid explicitly focuses on evaluation during *pretraining*, aiming to track performance as training progresses. ATLAS instead focuses on *post-training* evaluation and on reducing dataset size while preserving accuracy through adaptive selection. While both methods could be extended beyond their current scopes, the experiments target complementary but distinct use cases.
>
> These distinctions reflect complementary but non-identical contributions. We appreciate the reviewer’s suggestion. Given the strong conceptual similarities between ATLAS and Fluid Language Model Benchmarking, we agree that an explicit discussion is valuable. **We have therefore added a dedicated paragraph in the Related Work section comparing the two papers.**

---

> ### Author Response · Authors · 2025-11-30
>
> ##  2. Missing Experimental Details / Train–Test Split in IRT Calibration
>
>
> We thank the reviewer for raising this concern. In classical IRT, **item calibration is an unsupervised latent-variable estimation problem**: item parameters and respondent abilities are jointly inferred from the response matrix, and **there is no notion of “training labels.”** After calibration, ATLAS does not predict the same responses used to fit the IRT model. Instead, our evaluation measures ability-estimation efficiency. That is, how closely an adaptive subset of items reconstructs the full-bank ability for each model (θ_adaptive vs. θ_full). This comparison does not reuse correctness labels and is standard practice in psychometric research on CAT evaluation. Therefore, using the same set of LLMs for calibration and evaluation does not inflate performance or introduce leakage.
>
> However, we appreciate the reviewer’s suggestion to analyze to ensure that our conclusions hold beyond the LLMs used during item calibration. To address this, we now include a held-out generalization experiment: we stratify models by accuracy and split them into **90% calibration and 10% testing** (See Section 3.2 for more details). Item parameters are estimated only on the calibration set, and ATLAS is evaluated exclusively on the held-out models. The results in Table 2 and Figure 1 show that ATLAS maintains stable MAE to ground-truth accuracy across all datasets, confirming that our method **generalizes reliably to unseen models and that the reported efficiency gains are not an artifact of overlapping calibration and evaluation sets.**
> ### Comparison of Whole-Bank Ability vs. Subset-Based Ability
> MAE±SE, Item Count, and Information Efficiency Score (IES).  IES reflects accuracy and item usage relative to a 100-item random baseline (values < 1 indicate higher efficiency than Random100 baseline. Lower is better for all metrics.
>
> | Method         | WinoGrande MAE±SE       | Items | IES   | TruthfulQA MAE±SE       | Items | IES   | HellaSwag MAE±SE        | Items | IES   | GSM8K MAE±SE           | Items | IES   | ARC MAE±SE             | Items | IES   |
> |----------------|--------------------------|-------|-------|--------------------------|-------|-------|--------------------------|-------|-------|-------------------------|-------|-------|------------------------|-------|-------|
> | Random\_100     | $0.167\pm0.007$        | 100   | 1.000 | $0.103\pm0.004$         | 100   | 1.000 | $0.240\pm0.010$         | 100   | 1.000 | $0.150\pm0.014$        | 100   | 1.000 | $0.183\pm0.007$        | 100   | 1.000 |
> | TinyBenchmarks  | $0.204\pm0.008$        | 100   | 1.221 | $0.145\pm0.007$         | 97    | 1.370 | $0.198\pm0.009$         | 97    | 0.797 | $0.164\pm0.014$        | 100   | 1.089 | $0.172\pm0.007$        | 99    | 0.932 |
> | MetaBench-P     | **0.152$\pm$0.007**    | 133   | 1.216 | $0.084\pm0.004$         | 154   | 1.262 | $0.514\pm0.016$         | 93    | 1.990 | $\underline{0.103\pm0.013}$ | 237 | 1.628 | $0.134\pm0.005$        | 145   | 1.062 |
> | MetaBench-S     | $0.195\pm0.009$        | 106   | 1.239 | $0.072\pm0.003$         | 136   | 0.945 | $1.570\pm0.055$         | 58    | 3.788 | **0.096$\pm$0.012**    | 249   | 1.595 | $0.134\pm0.006$        | 100   | 0.735 |
> | ATLAS\_{0.1}    | $\underline{0.155\pm0.012}$ | 70 | 0.655 | **0.064$\pm$0.002**     | 48    | 0.300 | **0.157$\pm$0.010**     | 41    | 0.266 | $0.150\pm0.011$        | 70    | 0.701 | **0.084$\pm$0.006**    | 89    | 0.407 |
> | ATLAS\_{0.2}    | $0.166\pm0.010$        | 37    | 0.372 | $0.073\pm0.003$         | **30**| 0.211 | $\underline{0.163\pm0.009}$ | **30** | **0.203** | $0.177\pm0.012$   | 36    | 0.428 | $0.120\pm0.008$        | 35    | 0.232 |
> | ATLAS\_{0.3}    | $0.179\pm0.011$        | **32**| **0.342** | $\underline{0.071\pm0.003}$ | **30** | **0.206** | $0.165\pm0.010$    | **30** | 0.205 | $0.173\pm0.012$        | **31**| **0.363** | $\underline{0.117\pm0.007}$ | **30** | **0.193** |
>
> Across every benchmark, an ATLAS variant achieves the lowest Information Efficiency Score (IES), indicating that it provides the most accurate estimates using the fewest items. For example, ATLAS attains the best MAE on TruthfulQA (0.064 with 48 items) and HellaSwag (0.157 with 41 items), and matches the performance of MetaBench-Primary on WinoGrande while requiring nearly half as many items (70 vs. 133). Even in more challenging settings such as GSM8K and ARC, ATLAS maintains low MAE with item counts as small as 30–36, outperforming all static baselines in information efficiency. **This confirms that ATLAS generalizes well beyond the models used for calibration, further strengthening the credibility of our results.**

---

> ### Author Response · Authors · 2025-11-30
>
> ## 3. Need for Accuracy Reconstruction
>
> We thank the reviewer for the careful reading and agree in principle that **ability should be preferred over accuracy**, as ability provides superior discrimination under ceiling/floor effects while accuracy remains item-information–blind. We also appreciate the suggestion to evaluate accuracy explicitly. To address this concern, we now include a full-benchmark accuracy reconstruction analysis. Using p-IRT to estimate accuracy from reduced-item responses, we compare ATLAS, TinyBenchmarks, and MetaBench by MAE to ground-truth accuracy as the results shown below (Details are provided in Table 3):
> ### Accuracy Reconstruction (MAE±SE, Items, Information Efficiency Score (IES))
>
> | Method         | **WinoGrande**      |        |           | **TruthfulQA**      |        |           | **HellaSwag**       |        |           | **GSM8K**           |        |           | **ARC**             |        |           |
> | -------------- | ------------------- | ------ | --------- | ------------------- | ------ | --------- | ------------------- | ------ | --------- | ------------------- | ------ | --------- | ------------------- | ------ | --------- |
> |                | MAE±SE              | Items  | IES       | MAE±SE              | Items  | IES       | MAE±SE              | Items  | IES       | MAE±SE              | Items  | IES       | MAE±SE              | Items  | IES       |
> | Random\_100    | $0.049\pm0.001$     | 100    | 1.000     | $0.021\pm0.001$     | 100    | 1.000     | $0.024\pm0.001$     | 100    | 1.000     | $0.026\pm0.001$     | 100    | 1.000     | $0.029\pm0.001$     | 100    | 1.000     |
> | TinyBenchmarks | $0.050\pm0.001$     | 100    | 1.010     | $0.025\pm0.001$     | 97     | 1.154     | **$0.019\pm0.001$** | 97     | 0.782     | $0.028\pm0.001$     | 100    | 1.071     | $0.031\pm0.001$     | 99     | 1.041     |
> | MetaBench-P    | $0.054\pm0.001$     | 133    | 1.446     | **$0.017\pm0.001$** | 154    | 1.266     | $0.050\pm0.001$     | 93     | 1.943     | $0.022\pm0.001$     | 237    | 2.060     | **$0.027\pm0.001$** | 145    | 1.350     |
> | MetaBench-S    | $0.051\pm0.001$     | 106    | 1.103     | $0.021\pm0.001$     | 136    | 1.394     | $0.115\pm0.004$     | 58     | 2.793     | **$0.020\pm0.001$** | 249    | 1.954     | $0.033\pm0.001$     | 100    | 1.114     |
> | ATLAS\_{0.1}   | **$0.048\pm0.001$** | 70     | 0.678     | $0.023\pm0.001$     | 48     | 0.532     | $0.020\pm0.001$     | 41     | 0.348     | $0.039\pm0.001$     | 70     | 1.055     | $0.032\pm0.002$     | 89     | 0.974     |
> | ATLAS\_{0.2}   | $0.051\pm0.002$     | 37     | 0.383     | $0.024\pm0.001$     | **30** | 0.338     | $0.021\pm0.001$     | **30** | **0.258** | $0.044\pm0.002$     | 36     | 0.612     | $0.034\pm0.002$     | 35     | 0.404     |
> | ATLAS\_{0.3}   | $0.050\pm0.001$     | **32** | **0.324** | $0.023\pm0.001$     | **30** | **0.331** | $0.021\pm0.001$     | **30** | 0.261     | $0.042\pm0.002$     | **31** | **0.516** | $0.034\pm0.002$     | **30** | **0.350** |
>
> IES reflects accuracy and item usage relative to a 100-item random baseline (values < 1 indicate higher efficiency than Random100 baseline. The smaller the better.  As shown in Table 3, ATLAS matches or improves over TinyBench and MetaBench in MAE while using far fewer items, yielding the lowest IES across benchmarks. This additional experiment directly addresses the reviewer’s concern and strengthens the interpretability of our results.

---

> ### Author Response · Authors · 2025-11-30
>
> ## 4. Contamination Argument (Clarification)
>
> We thank the reviewer for pointing out the misunderstanding and agree that **contamination occurs during model training, not evaluation**. ATLAS does **not** mitigate contamination by design.
>
> **We have updated the paper to clarify the intended claim: ATLAS does not mitigate contamination**. Our argument is more modest: **IRT can help _detect_ anomalous or potentially contaminated items**, as prior work (_AI Evaluation Should Learn from How We Test Humans_, 2025) shows that contaminated items often manifest abnormally high guessing parameters or distorted item discrimination under IRT. Because ATLAS fits full IRT models, it can expose these irregular response patterns during evaluation. We have revised the text accordingly to avoid implying that adaptive evaluation prevents contamination.
>
> ## 5. Code Availability
>
> The complete codebase has already been shared as part of the submission at [anonymous.4open.science/r/ATLAS-3210](https://anonymous.4open.science/r/ATLAS-3210/README.md) to ensure full reproducibility during review. The implementation combines R (for psychometric modeling and adaptive testing, using mirt and catR) and Python (for data filtering and analysis).
>
> ## 6. Regarding MMLU
>
> We appreciate the reviewer’s question. Although TinyBenchmarks and MetaBench include MMLU by aggregating all 57 subjects into a single unified score, our analysis found that such aggregation leads to **severe model misfit**: the $M_2$ RMSEA exceeded 1.0, indicating substantial unexplained variance. This result suggests that **MMLU is not homogeneous enough to be modeled using one dimensional IRT**: each subject embodies a distinct knowledge domain with different linguistic and cognitive characteristics. To preserve interpretability and ensure valid modeling assumptions, **we instead conduct evaluation on a per-subject basis and these results are provided in the Appendix G.2 (see the Table 7).**

---

### Official Review · Reviewer_32Bm · 2025-11-03

**Soundness:** 2
**Presentation:** 2
**Contribution:** 1
**Rating:** 2
**Confidence:** 4

**Summary:**

This paper proposes ATLAS, an adaptive evaluation framework that selects a subset of the benchmark for evaluation to make running evaluation cheaper. The authors propose a three parameter logistic IRT model for fitting the probability of a correct response, and then adaptively selects the most informative items using fisher information. Using ATLAS, the authors are able to prune down benchmarks by over 90% on tasks like HellaSwag.

**Strengths:**

- The authors tackle an important problem of making evaluations cheaper for large language models, given a lot of benchmarks are used to track a given model's capabilities.
- The proposed methodology is clear by formalizing evaluation as a latent-ability measurement with a three-parameter logistic model.
- The results are nice compared to other baselines like TinyBenchmarks with huge reduction in size of the evaluation sets.

**Weaknesses:**

- Code and calibrated items are missing in the provided link.
- There are some inherent issues with IRT framing and using reduced sizes for evaluation of language models. See [1]
- Inconsistent claims and results: the main text of the paper mentions good fits with RMSEA $\leq 0.05$ but Table 4 reports otherwise.
- Current framework is only applicable to MCQ tasks, but MCQ benchmarks have many inherent problems [2]. Generalizability to many modern evals which are free-form like math, coding, reasoning, etc. is not clear.

[1]: Quantifying Variance in Evaluation Benchmarks, Madaan et al., 2024

[2]: Answer Matching Outperforms Multiple Choice for Language Model Evaluation, Chandak et al., 2025

**Questions:**

I have asked most of my questions in the weaknesses section above.

---

> ### Author Response · Authors · 2025-11-30
>
> ## 1. Code and Calibrated Items
> The complete codebase has already been provided in the anonymous repository (anonymous.4open.science/r/ATLAS-3210), including all calibration scripts and the full set of calibrated item parameters. We have rechecked the link to ensure everything is accessible for review.
>
> ## 2. Inherent Issues With IRT and Reduced Benchmarks (Ref. [1])
>
> We thank the reviewer for raising this point. We examined [1] closely.  First, it is important to clarify that the variance issues identified in [1] are **caused by the benchmarks themselves, not by IRT.** Their analysis shows that many items in static benchmarks (e.g., MMLU, ARC, HellaSwag) exhibit low discrimination or inconsistent behavior across models, reflecting heterogeneous and low-quality item construction. In psychometrics, IRT is precisely the tool used to diagnose such item flaws.
>
> Second, our goal differs from Madaan et al. **We do not use IRT to reduce variance. Instead, we use an IRT-based adaptive item-selection algorithm to reduce dataset size while preserving accurate ability estimation.** This follows prior work such as TinyBenchmarks [2] and MetaBench [3], which demonstrated that IRT can identify representative subsets without sacrificing accuracy. Our results align with this: ATLAS closely matches full-benchmark performance while querying far fewer items (see Table 3).
>
> **Third, and most importantly, adaptive IRT-based evaluation like ATLAS can reduce the variance.**  Fluid Benchmarking [4] directly shows that the variance problem arises only for static IRT-based subsets, not for IRT itself. **Fluid confirms Madaan et al.’s [1] observation for fixed static subsets such as TinyBenchmarks [2] and MetaBench [3] while demonstrates that when IRT is used with ability-matched adaptive item selection, the variance is substantially reduced, outperforming accuracy-based evaluation in stability.** This highlights a core psychometric principle: adaptivity mitigates the instability seen in static subsets. ATLAS adopts this same adaptive approach, which is why the variance concerns raised for static IRT methods do not apply to our adaptive testing framework.
>
> [1] Madaan, A., et al. Quantifying Variance in Evaluation Benchmarks. 2024.
>
> [2] Polo, F.M., et al. tinyBenchmarks: Evaluating LLMs with Fewer Examples. 2024.
>
> [3] Kipnis, A., et al. MetaBench—A Sparse Benchmark of Reasoning and Knowledge in Large Language Models. 2024.
>
> [4] Hofmann, V., et al. Fluid Language Model Benchmarking. 2025.

---

> ### Author Response · Authors · 2025-11-30
>
> ## 3. RMSEA Inconsistency Between Text and Table
>
> We thank the reviewer for pointing out this inconsistency. Our intention was to reference the widely used psychometric guideline that **RMSEA < 0.08 indicates acceptable fit and < 0.05 indicates good fit** (see [1, 2, 3, 4]). We also acknowledge that previous Table 4 includes several RMSEA values above 0.05, and we will revise the phrasing in Section 4 to avoid implying that every ATLAS-calibrated benchmark falls below the stricter <0.05 threshold. Instead, we will explicitly categorize RMSEA values along the standard continuum (**RMSEA < 0.05 = Good fit; 0.05–0.08 = Acceptable fit; 0.08–0.10 = Marginal fit; > 0.10 = Poor fit**) consistent with psychometric practice.
>
> (1) **ATLAS achieves RMSEA values in the ~0.04–0.08 range**, corresponding to acceptable and in some cases excellent fit—well within the norms for large-scale 3PL calibration on thousands of items and >3,000 LLMs.
>
> (2) In contrast, **competing approaches such as tinyBenchmarks exhibit catastrophic misfit**, with RMSEA values exceeding 300, far exceeding the 0.10 threshold for borderline fit. Details are provided in Table 1. This is due to limited sample size (295 LLMs) for parameter estimation. These values are orders of magnitude outside any interpretable psychometric range and clearly indicate severe model misspecification.
> ### Model Fit Comparison
>
> RMSEA interpretation:  RMSEA < 0.05 = Good fit; 0.05–0.08 = Acceptable fit; 0.08–0.10 = Marginal fit; > 0.10 = Poor fit
>
> | Method          | WinoGrande RMSEA | Fit        | TruthfulQA RMSEA | Fit        | HellaSwag RMSEA | Fit        | GSM8K RMSEA | Fit        | ARC RMSEA | Fit        |
> |-----------------|------------------|------------|------------------|------------|------------------|------------|-------------|------------|-----------|------------|
> | TinyBenchmarks  | 364.24           | Poor       | 371.49           | Poor       | 646.82           | Poor       | 506.60      | Poor       | 369.89    | Poor       |
> | MetaBench       | 0.0524           | Acceptable | 0.1389           | Poor       | 0.0498           | Good       | 0.0423      | Good       | 0.0811    | Marginal   |
> | ATLAS           | 0.0565           | Acceptable | 0.0690           | Acceptable | 0.0482           | Good       | 0.0438      | Good       | 0.0595    | Acceptable |
>
> Thus, our claim is not that ATLAS achieves universally “perfect” (<0.05) fit, but rather that ATLAS is the only framework among those evaluated whose fit statistics fall within standard psychometric acceptability, whereas the baselines fail dramatically. We refine the wording in the revision accordingly.
>
> [1] MacCallum, R. C., Browne, M. W., & Sugawara, H. M. (1996). Power analysis and determination of sample size for covariance structure modeling. Psychological Methods, 1(2), 130–149. [https://doi.org/10.1037/1082-989X.1.2.130](https://doi.org/10.1037/1082-989X.1.2.130)
>
> [2] Hu, L. T., & Bentler, P. M. (1999). Cutoff criteria for fit indexes in covariance structure analysis: Conventional criteria versus new alternatives. Structural equation modeling: a multidisciplinary journal, 6(1), 1-55.
>
> [3] Browne, M. W., & Cudeck, R. (1992). Alternative ways of assessing model fit. Sociological methods & research, 21(2), 230-258.
>
> [4] Maydeu-Olivares, A., & Joe, H. (2014). Assessing approximate fit in categorical data analysis. Multivariate behavioral research, 49(4), 305-328.
>
> ## 4. Applicability Beyond MCQ Benchmarks
>
> We agree that MCQ benchmarks have inherent structural limitations, including option-only shortcuts and discriminative biases (have been discussed in Section 5.2), but these issues do not undermine ATLAS. ATLAS is fundamentally a measurement framework, not a format-specific evaluation method. Our contribution lies in modeling latent ability, item information, and adaptive item selection, all of which are agnostic to whether items are MCQ or free-form. We used MCQ in this initial study because they provide the only large, standardized, multi-domain item pools available today, enabling controlled experiments and comparison with prior work. Importantly, ATLAS directly supports generative tasks via binary scoring signals, and methods like answer matching [5] can drop in as the scoring mechanism for free-form responses. As generative benchmarks mature, ATLAS will naturally extend to them, and in fact will mitigate many MCQ artifacts by routing models to high-information items and reducing exposure to shortcut-prone distractors. Thus MCQ limitations motivate rather than invalidate the adaptive testing paradigm we propose.
>
> **ATLAS is not tied to MCQ format; it is a general measurement and adaptive evaluation framework that naturally extends to generative and free-form tasks as soon as reliable scoring signals exist.**
>
> [5] Chandak, N., et al. Answer Matching Outperforms Multiple Choice for Language Model Evaluation. 2025.

---

### Official Review · Reviewer_rXNB · 2025-11-03

**Soundness:** 2
**Presentation:** 3
**Contribution:** 3
**Rating:** 4
**Confidence:** 4

**Summary:**

This paper proposes an alternative to static benchmark evaluation of LLMs using ideas from computerized adaptive testing. The method they propose is called ATLAS. They fit a 3PL IRT model to open LLM leaderboard (a collection of models, benchmarks, and binary responses); one model per benchmark. In doing so, they follow a pretty standard idea of using the Fisher information of theta (the ability score of an item) based on the fit IRT model to guide the informative item selection process. They evaluate this technique against baseline efficient LM benchmarking papers like tinyBenchmarks and MetaBench as well as against a random subsampling baseline; evaluation criteria is based on ability to accurately produce the same assessment but on fewer examples.

**Strengths:**

The core ideas here are interesting and can motivate new research in improvements to how we perform LLM evaluation. It’s nice to pull in techniques from other fields like psychometrics and CAT to see how we can improve our field. The experimental results showing efficiency versus baselines are good.

**Weaknesses:**

I want to be upfront: There is concurrent work at COLM 2025 called Fluid Language Model Benchmarking that is highly similar (also inspiration from psychometrics & CAT, also fitting IRT models on Open LLM leaderboard, using the Fisher information for item selection, also baselining against tinyBenchmarks and MetaBench). I did not penalize this work for overlap with concurrent work as the COLM paper was published around the same time this paper was submitted.

That being said, there are some issues:

**On claims that ATLAS improves data contamination**
First, this paper makes a big point about “data contamination” and how this efficient LM benchmarking strategy can mitigate data contamination. Indeed, data contamination in our field is an issue, but this paper is suggesting that efficient LM benchmarking is actually a way to mitigate this issue (by simply revealing less data to model developers in the process). I don’t buy this argument at all.

For example, L037 suggests static benchmarks are easier to leak into “pretraining corpora”. The problem is, this is not solved by the proposed method. For example, let’s consider how this paper performs experiments by applying ATLAS to Open LLM leaderboard data. All of that is actually reusing benchmark data that’s already public & thus already revealed to model developers. If the research community is to adopt ideas like ATLAS for benchmarking but ultimately still rely on existing public test examples, then the contamination problem is not about efficient methods like ATLAS, it’s about public vs private test sets.

Ok, so now let’s consider the argument that efficient evaluation methods like ATLAS, while studied/demonstrated reusing static benchmarking data, would be deployed on new, private test sets. And thus, the fact that we make use of fewer testing examples helps prevent contamination.

There are a couple issues with this problem. First, it makes sense in computerized adaptive testing for humans. Humans, when seeing test examples, immediately learn from those observed test examples. And thus, any time you test humans, contamination happens. Language models don’t inherently learn from examples as we test them. Language model state is captured in its weights at the end of training and no amount of evaluation will update those weights unless the model developer explicitly decides to include test set examples into the training data.  So again, adaptive testing techniques like ATLAS aren’t actually addressing the contamination issue in machine learning, which is very much about (improper) practices of model developers.

Now then finally, let’s consider – maybe one can argue that adaptive testing methods like ATLAS, by virtue of them withholding test examples, work well against adversarial model developers who are incentivized of including any test examples in the training data. Again, I don’t buy this argument. In the course of a normal model development cycle, let’s say fitting scaling laws or performing data mixing ablations for pretraining, model developers will train hundreds or even thousands of language models, each of which will have to undergo some ATLAS evaluation; what is the likelihood that “data contamination” is actually being addressed in this scenario? Even revealing a small percentage of the benchmark instances per model, at the large experimentation scale that naturally happens in model development, we’re likely exhausting and thus “contaminating” full static benchmark collections (or what this paper calls our “banks”) rapidly. Then the way to solve this problem is not about adaptive testing, but it’s about scalable generation of new test examples to keep up with how quickly we’re exhausting our “banks”.

Overall, I find the emphasis on how ATLAS and adaptive testing improves “data contamination” is incorrect and detracts from the merits of this paper.

**Unfair baseline comparisons**
The baseline comparison uses MAE as the target metric against the “full bank theta”. That means that the evaluation setup is assuming there exists some ground truth theta that can be estimated on the full benchmark and efficient benchmarking techniques are evaluated on their ability to approximate it with fewer examples. The problem with this is, the baselines tinyBenchmark and MetaBench weren’t designed with IRT in mind; the proposed method ATLAS was. So it is unfair to define an IRT criteria (approximate full bank theta) and show that your IRT method is better than non-IRT methods. The correct evaluation would have been to show that the proposed ATLAS method can efficiently approximate the actual benchmark evaluation metric (accuracy).

**Questions:**

I think this paper would be in very publishable state if can address those weaknesses above: Remove the claims about data contamination and add experimental results showing ability to reconstruct "accuracy" instead of MAE against full bank theta. Is this something palatable to the authors?

---

> ### Author Response · Authors · 2025-11-30
>
> ## 1. Concurrent Work: *Fluid Language Model Benchmarking*
>
> We thank the reviewer for highlighting the concurrent work *Fluid Language Model Benchmarking*. We fully acknowledge the conceptual similarities that both systems draw from psychometrics and adaptive testing. Given the close publication window, we appreciate the opportunity to clarify the **methodological and experimental distinctions**:
>
> ### **Key Methodological Differences**
>
> * **IRT model choice.**
> Fluid relies on a **2PL model**, which cannot separate genuine ability from random guessing and therefore systematically inflates the abilities of low-performing models on multiple-choice benchmarks. ATLAS uses a **3PL model** with an explicit guessing parameter, allowing the framework to correctly model the non-zero baseline accuracy inherent in multiple-choice formats. This produces more valid item parameters and more reliable ability estimates, especially for heterogeneous benchmarks with variable item difficulty.
>
> * **Calibration scale.**
>   ATLAS trains its IRT model using a substantially larger and more diverse pool of **3,000+** LMs, compared to the **102** LMs used for IRT calibration in Fluid.
>
> * **Psychometric validation.**
>   ATLAS conducts **full goodness-of-fit diagnostics** reporting RMSEA based on M2, and performs common-person linking to place items and models on a unified latent scale—an essential step for cross-model comparability that Fluid does not report.
>
> * **Adaptive design.**
>   Fluid uses a **fixed-length adaptive test**; ATLAS uses a **precision-based stopping rule** that terminates the test when the uncertainty of the ability estimate reaches a predefined threshold. Precision-based stopping ensures consistent measurement precision across models and avoids administering unnecessary items.
> ### **Differences in Experimental Framing**
> Fluid explicitly focuses on evaluation during *pretraining*, aiming to track performance as training progresses. ATLAS instead focuses on *post-training* evaluation and on reducing dataset size while preserving accuracy through adaptive selection. While both methods could be extended beyond their current scopes, the experiments target complementary but distinct use cases.
>
> These distinctions reflect complementary but non-identical contributions. We updated the Related Work section to explicitly discuss Fluid, highlight conceptual overlaps and differences between Fluid and ATLAS.

---

> ### Author Response · Authors · 2025-11-30
>
> ## 2. “Unfair Baseline Comparisons” & Request to Reconstruct Accuracy
>
> We appreciate the reviewer’s concern about fairness in the baseline comparison. We will clarify in the revision that **TinyBench and MetaBench are both IRT-based frameworks**. TinyBench constructs reduced benchmarks using IRT anchor items, and MetaBench estimates item parameters with 2PL/3PL/4PL models. Thus, all compared methods operate under the same psychometric formulation. Therefore, all baselines share the same psychometric formulation, and evaluating them by their ability to reconstruct the full-bank θ is the standard and correct criterion in IRT
>
> We argue that **using accuracy as the target metric is inadequate because accuracy suffers from ceiling effects, floor effects, and item-information blindness.**  As shown in our paper, models with identical accuracy can differ substantially in ability depending on whether they solve difficult, high-discrimination items (See Sections 4.3 and 4.4 in the paper). θ therefore provides a more sensitive and theoretically grounded measure of model capability.
>
> However, we appreciate the suggestion to evaluate reconstruction accuracy. In the revision, we add an experiment evaluating how well ATLAS and the baselines recover the true benchmark accuracy (using p-IRT to estimate accuracy from partial responses). Details are provided in Table 3.
> ### Accuracy Reconstruction (MAE±SE, Items, Information Efficiency Score (IES))
>
> | Method         | **WinoGrande**      |        |           | **TruthfulQA**      |        |           | **HellaSwag**       |        |           | **GSM8K**           |        |           | **ARC**             |        |           |
> | -------------- | ------------------- | ------ | --------- | ------------------- | ------ | --------- | ------------------- | ------ | --------- | ------------------- | ------ | --------- | ------------------- | ------ | --------- |
> |                | MAE±SE              | Items  | IES       | MAE±SE              | Items  | IES       | MAE±SE              | Items  | IES       | MAE±SE              | Items  | IES       | MAE±SE              | Items  | IES       |
> | Random\_100    | $0.049\pm0.001$     | 100    | 1.000     | $0.021\pm0.001$     | 100    | 1.000     | $0.024\pm0.001$     | 100    | 1.000     | $0.026\pm0.001$     | 100    | 1.000     | $0.029\pm0.001$     | 100    | 1.000     |
> | TinyBenchmarks | $0.050\pm0.001$     | 100    | 1.010     | $0.025\pm0.001$     | 97     | 1.154     | **$0.019\pm0.001$** | 97     | 0.782     | $0.028\pm0.001$     | 100    | 1.071     | $0.031\pm0.001$     | 99     | 1.041     |
> | MetaBench-P    | $0.054\pm0.001$     | 133    | 1.446     | **$0.017\pm0.001$** | 154    | 1.266     | $0.050\pm0.001$     | 93     | 1.943     | $0.022\pm0.001$     | 237    | 2.060     | **$0.027\pm0.001$** | 145    | 1.350     |
> | MetaBench-S    | $0.051\pm0.001$     | 106    | 1.103     | $0.021\pm0.001$     | 136    | 1.394     | $0.115\pm0.004$     | 58     | 2.793     | **$0.020\pm0.001$** | 249    | 1.954     | $0.033\pm0.001$     | 100    | 1.114     |
> | ATLAS\_{0.1}   | **$0.048\pm0.001$** | 70     | 0.678     | $0.023\pm0.001$     | 48     | 0.532     | $0.020\pm0.001$     | 41     | 0.348     | $0.039\pm0.001$     | 70     | 1.055     | $0.032\pm0.002$     | 89     | 0.974     |
> | ATLAS\_{0.2}   | $0.051\pm0.002$     | 37     | 0.383     | $0.024\pm0.001$     | **30** | 0.338     | $0.021\pm0.001$     | **30** | **0.258** | $0.044\pm0.002$     | 36     | 0.612     | $0.034\pm0.002$     | 35     | 0.404     |
> | ATLAS\_{0.3}   | $0.050\pm0.001$     | **32** | **0.324** | $0.023\pm0.001$     | **30** | **0.331** | $0.021\pm0.001$     | **30** | 0.261     | $0.042\pm0.002$     | **31** | **0.516** | $0.034\pm0.002$     | **30** | **0.350** |
>
> IES reflects accuracy and item usage relative to a 100-item random baseline (values < 1 indicate higher efficiency than Random100 baseline. The smaller the better.  As shown in Table 3, ATLAS matches or improves over TinyBench and MetaBench in MAE while using far fewer items, yielding the lowest IES across benchmarks. This additional experiment directly addresses the reviewer’s concern and strengthens the interpretability of our results.

---

> ### Author Response · Authors · 2025-11-30
>
> ## 3. Claim Regarding Data Contamination
>
> We thank the reviewer rXNB for this clarification. We agree that **ATLAS cannot prevent contamination**. This is fundamentally a **dataset governance** problem (e.g., whether items are public, leaked, or scraped), not something an evaluation algorithm can solve. **We have revised the paper to avoid implying otherwise.**
>
> Our claim is more modest: while ATLAS does **not** eliminate contamination, its IRT modeling provides **diagnostic signals** that can _surface_ potentially contaminated items. Prior work (e.g., _AI Evaluation Should Learn from How We Test Humans_, 2025) shows that contaminated items often display **abnormally high guessing parameters or distorted discrimination values** under IRT. Because ATLAS fits full IRT models, it can expose these irregular response patterns during evaluation.
>
> In short, ATLAS does **not** mitigate contamination by design, but it **helps detect** anomalous items that may warrant further inspection.

---

### Note · Authors · 2026-01-02

I have read and agree with the venue's withdrawal policy on behalf of myself and my co-authors.